

# Climate change risks illustrated by the IPCC "burning embers"

Philippe Marbaix[1], Alexandre K. Magnan[2,3,4], Veruska Muccione[5], Peter W. Thorne[6], Zinta Zommers[7]

[1] Université catholique de Louvain, Louvain-la-Neuve, Belgium.
[2] Cawthron Institute, Nelson, New Zealand.
[3] Institute for Sustainable Development and International Relations, Sciences-Po, Paris, France.
[4] World Adaptation Science Programme, United Nations Environment Programme (Secretariat), Nairobi, Kenya.
[5] Swiss Federal Research Institute WSL, Birmensdorf, Switzerland
[6] ICARUS Climate Research Centre, Maynooth University, Maynooth, Ireland
[7] United Nations Office for the Coordination of Humanitarian Affairs, New York, United States

*Correspondence to*: Philippe Marbaix (philippe.marbaix@uclouvain.be)

**Abstract.** The completion of the Sixth Assessment Cycle of the Intergovernmental Panel on Climate Change (IPCC) provides a unique opportunity to understand where the world stands on climate change-related risks to natural and human systems, at the global level as well as for specific regions and sectors. Since its Third Assessment Report, released two decades ago, the IPCC has developed a synthetic representation of how risks increase with global warming, known as "burning embers"
diagrams due to the colours used. While initially designed to illustrate five overarching Reasons for Concern, these diagrams have been progressively applied to risks in specific systems and regions over the last 10 years. However, the information gathered through expert elicitation and the resulting quantitative risk assessments have hitherto remained scattered within and across reports and specific data files. This paper overcomes this limitation by developing an ember database and an associated online "climate risks ember explorer" to facilitate the exploration of the assessed risks. The data are also available in an archive
file in a widely accessible format (doi:10.5281/zenodo.12626977, Marbaix et al. 2024). Important aspects of data homogenisation are discussed, and an approach to structuring information on assessed risk increases is presented. Potential uses of the data are explored through aggregated analyses of risks and adaptation benefits, which show that, excluding high adaptation cases, half of the assessed risks levels increase from a moderate to a high risk between 1.5°C and 2 to 2.3°C of global warming, a result which is consistent with the separate assessment of the Reasons for Concern by the IPCC. The
database lays the groundwork for future risk assessments and the development of burning embers by providing a standardised baseline of risk data. It also highlights important areas for improvement in the forthcoming IPCC Seventh Assessment Cycle, in particular towards systematic, homogenous, and structured collection of information on illustrated risk increases, a comprehensive coverage of impacted regions, a systematic consideration of adaptation and/or vulnerability levels, and possibly the coverage of risks from response measures. In the context of an ever-growing literature and knowledge, the facility described
herein has the potential to help in synthesising and illustrating risks across scales and systems in a more consistent and comprehensive way.





# 1   Introduction

Since its creation in 1988, the Intergovernmental Panel on Climate Change (IPCC) has been tasked to regularly synthesise and
assess the scientific literature on anthropogenic climate change causes, processes, impacts on ecosystems and socio-economic
consequences as well as possible responses (IPCC, 1989; UN General Assembly, 1988). The synthesis work has raised
important methodological challenges related to the wide variety of information provided by various disciplines, with multiple
uncertainties leading to differences among expert judgments (Preface and SPM in Watson et al., 1996). Risk analyses are
heterogeneous in terms of the metrics used, the risk processes assessed, the natural and/or human systems analysed, and both
the spatial and the temporal scales considered. Despite these difficulties, having a synthetic view of risks is particularly
important to inform decision-making. This was particularly the case to help define what might constitute a "dangerous
anthropogenic interference with the climate system", which the Framework Convention aims to avoid according to its Article
2 (UNFCCC, 1992). With this objective in mind, the IPCC devoted a chapter of its 3rd assessment report (Smith et al., 2001)
to bringing together scientific knowledge that could provide a global overview of the risks, by creating a new concept: the «
Reasons for Concern » (RFCs). These are divided into five topics: unique and vulnerable systems, extreme events, distribution
of impacts, aggregated impacts and large-scale singular events. For each of these concerns, the increase in risk as a function
of temperature was presented using a colour scale illustrating the levels of risk (Ahmad et al., 2001; Smith et al., 2001). The
colours chosen, from white to red, have given these diagrams the nickname of "burning embers".

The Reasons for Concern have been re-assessed in each subsequent IPCC report. Starting with the 5th assessment report (AR5),
the burning ember diagrams have been applied to more specific risks (IPCC, 2014b). AR5 also introduced an extended risk
scale with four discrete risk levels (Undetectable, Moderate, High, Very high, the latter shown in purple; table 1, (Zommers et
al., 2020)). At the same time, the confidence in the assessment of each risk transition (the levels of warming at which risk
increases from one level to the next) began to be assessed and reported (IPCC, 2014a; O'Neill et al., 2017a). Following standard
IPCC practice, a high level of confidence indicates that robust evidence is available and that there is high agreement about the
findings (Mastrandrea et al., 2010; Rawshan Ara Begum et al., 2022).



| Risk level | Definition | Colour coding in the burning embers |
|---|---|---|
| Undetectable | No associated impacts are detectable and attributable to climate change. | White |
| Undetectable to moderate | Intermediary (risk transition) | White to yellow |
| Moderate | Associated impacts are both detectable and attributable to climate change with at least medium confidence, also accounting for the other specific criteria for key risks (*). | Yellow |
| Moderate to high | Intermediary (risk transition) | Yellow to red |
| High | Severe and widespread impacts that are judged to be high on one or more criteria for key risks (*). | Red |
| High to very high | Intermediary (risk transition) | Red to purple |
| Very high | Very high risk of severe impacts and the presence of significant irreversibility or the persistence of climate-related hazards, combined with limited ability to adapt due to the nature of the hazard or impacts/risks. | Purple |
| (*) Key risks refer to climate risks having the potential to lead to severe consequences, and are therefore relevant to the interpretation of "dangerous anthropogenic interference with the climate system" (Magnan et al., 2023). The criteria for assessing key risks include (O'Neill et al., 2022): Magnitude of the consequences (related to pervasiveness, degree of change, irreversibility, potential for thresholds, cascading effects to other systems) Likelihood of adverse consequences Temporality / persistence, Ability to respond to risk (including but not only through adaptation) | | |

**Table 1. The risk scale used in the IPCC reports since AR5 (adapted from O'Neill et al., 2022).**

Information on future risks remains scattered in the literature and therefore synthesis work remains a challenge, which initiatives such as the Inter-Sectoral Impact Model Comparison Project (ISIMIP) are working to reduce by establishing a
common analytical framework (Rosenzweig et al., 2017). The IPCC has developed its approach to synthesising risks in a number of ways that may help to make it more systematic. In AR4, it established criteria to define 'key' vulnerabilities (Schneider et al., 2007), which were later referred to as "key risks". AR5 and AR6 used this concept to identify the risks that need to be taken into account in assessing the RFCs (O'Neill et al., 2022; Zommers et al., 2020). In AR6, these criteria were used to select the risks illustrated in the burning embers diagrams of some of the chapters (see in particular Bednar-Friedl et
al., 2022a; Lawrence et al., 2022). While burning embers were based on literature reviews and expert judgements since the TAR, a dedicated structured expert elicitation process has been formalised more recently and progressively enacted (Zommers





et al., 2020). The protocol involved several rounds of individual assessments of how risks change with climate, followed by sharing the judgments and discussing within the group dedicated to assess a given climate risk.

More recently, the IPCC has progressively applied the burning ember approach to various scales, from global to regional and
local level. In the Special Report on Climate Change and Land (SRCCL), it started to differentiate its analysis according to socio-economic development trajectories (Hurlbert et al., 2019). Risks were assessed in the context of vulnerability, exposure and/or adaptation potential considered consistent with one of the "shared socio-economic scenarios" (SSPs), developed over the last decade (O'Neill et al., 2017b). The special report on Ocean and cryosphere (SROCC) assessed increasing risks due to sea-level rise within a framework similar (but not identical) to the burning embers; for the first time in this type of assessment,
it distinguished two levels of implementation of response measures, which may include managed retreat and/or adaptation. (IPCC, 2019; Oppenheimer et al., 2019).

The recent completion of the Sixth Assessment Cycle of the IPCC (AR6) offers the largest compilation of synthetic risk assessments in the form of burning embers to date (almost 90% of all embers built so far occurred within the AR6 cycle), which provides a unique opportunity to understand climate risk in a more cross-region and cross-sector way – although the
regional embers still only cover around half of the world. The semi-standardised expert elicitation method and the burning embers diagrams provide consistency in the risk assessment across scales, systems and sectors; it offers a new overview of climate-related risks, forming a solid basis for risk communication and further research. The AR6 also made progress in terms of including adaptation scenarios when assessing future risks, hence also providing an opportunity to start understanding the potential role of adaptation efforts in terms of risk reduction at the global level (Magnan et al., 2021). In the end, all this
material is seen as important to feed into discussions under the United Framework Convention on Climate Change (UNFCCC), and AR6 outcomes had been widely recognized as an important contribution to the First Global Stocktake (UNFCCC, 2023).

While recent papers have made advances comparing risk and embers across different systems (Magnan et al., 2021), this paper goes a step further by considering all burning embers developed over the whole AR6 cycle. It proposes to structure a database (Sect. 2) to gather all this material in a harmonised collection of information, facilitating and improving access to the results
of past —and possibly future— burning ember expert elicitation efforts. Based on this, it illustrates how such a structured database can be used to analyse climate risk across scales and systems, highlighting the potential role of mitigation and adaptation (Sect. 3), and discusses possible contributions to future risk assessments and the communication of their results (Sect. 4).



## 2 A database of climate risks illustrated as «burning embers»

### 2.1 Objectives and structure of the database

#### 2.1.1 Presentation of ember data

The information associated with a given burning ember can be divided into three categories: descriptive information about the risk being considered, (semi-) quantitative estimates of how risk increases with the level of climate change and the associated confidence levels, and metadata including textual arguments for risk transitions. Figure 1 illustrates, how the information associated with a given burning ember is presented in an online interface, as an introduction to the content of the database. Ember data pages begin with the description of the risk under consideration. A table then provides the quantitative estimates of how risk increases with climate change and the associated confidence levels. This presentation is based on the practice in the Supplementary Materials of IPCC reports since the Special Report on global warming of 1.5°C (Hoegh-Guldberg et al., 2018; Zommers et al., 2020). The same structure is used in the input files for the "Ember Factory" software (Marbaix, 2020b), which was used to draw many of the AR6 embers. Finally, this view provides textual arguments for risk transitions and metadata. The database also contains information about how the embers are presented in figures, as explained below.

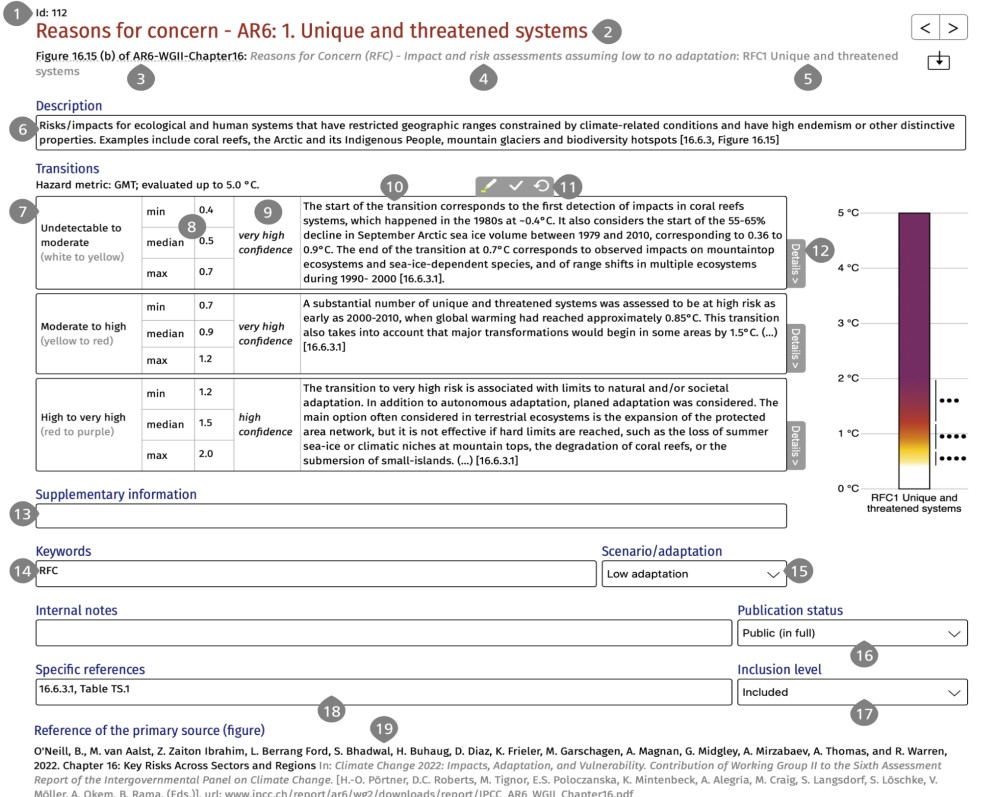



| | |
|---|---|
| 1 | Unique identifier of the ember in the database |
| 2 | "Long name" of the ember: defined by the editors of the database to clearly identify this ember among all others |
| 3 | Figure number, report, and chapter (clickable: links to the related page of the report on IPCC's website) |
| 4 | Ember group: the name of a set of embers in the report (it may be the title of a figure panel) |
| 5 | Ember name: as a rule, it is the label provided under the ember, in figures |
| 6 | Description of the risk assessed in this ember (verbatim from the report if available, otherwise synthetized by the editors of this database), with specific reference(s) to the source(s) within the reference provided at the bottom of the page |
| 7 | Transition name (standard wording used in IPCC reports, see table 1) |
| 8 | Hazard level (the "hazard" metric is usually global mean temperature change) at the beginning/median/end of the transition |
| 9 | Level of confidence in the assessment of the transition |
| 10 | Explanation: A summary of the elements explaining the assessment of the levels of climate change in each risk transition. As for [6], this summary can be verbatim from IPCC reports or not (this example was shortened for illustration) |
| 11 | Editors can add or change text within the standard viewing interface; these buttons are shown while editing |
| 12 | "Details" buttons are attached to each transition and may show additional text. Usage was limited and restricted to editors so far. |
| 13 | Details and clarifications related to the ember, such as elements of context or limitations of the assessment |
| 14 | Keywords may help in defining groups of 'related' embers for identification, illustration, or aggregated analysis |
| 15 | Adaptation level or scenario (optional; present when a risk is assessed for several scenarios) |
| 16 | Access to an ember may be restricted to editors while the data is being prepared |
| 17 | Inclusion level relates to API or archive file access: a few embers are listed as duplicates or irrelevant for analyses (see Table 3) |
| 18 &19 | References are provided in two ways: the main source of information is a chapter in an IPCC report [19]; in addition, editors may indicate details about the sections within reports and additional references if needed [18] |

**Figure 1: Example 'burning ember' presentation in the online interface of the database. Encoding by authorised editors is possible for the 'descriptive' text fields (2, 6, 10, 13, 14, 18). The assessed risks relate to a Reason for Concern (see introduction).**

### 110  2.1.2   Risk transitions

The key information to produce an ember diagram is the magnitude of climate change corresponding to the minimum (beginning) and maximum (ending) of each transition from one risk level to the next. In this database, the change against which risk is evaluated (driver) is called "hazard" – in line with IPCC glossaries (IPCC, 2022b). While most embers use global mean surface temperature change as a proxy for broader climate changes/hazards, a few embers relate to other metrics such as global

sea-level rise or $CO_2$ concentration (IPCC, 2014). For some embers, experts provided estimates of the median hazard within transitions, which defines the location of the 50% change in colour on the diagrams. Finally, a confidence level (low, medium, high, very high) is associated to each assessed transition, starting with the AR6 cycle (Mach et al., 2017).

Descriptions of the assessed risk(s) and explanations of the risk transitions are provided in the reports in different ways: some chapters explicitly link information on risks to each ember and the transitions it illustrates (for example Bednar-Friedl et al.,

2022b; Parmesan et al., 2022), while others assess risks and present embers in a more separated way (for example Cissé et al., 2022). The details of how and to what extent each risk transition is explained vary from chapter to chapter. As the information provided on a given risk can be long and spread across sections of a chapter without direct reference to the figure containing



the ember, it may not be easy to extract an accurate short description of the subject of each ember, or the reasons for the risk increase. Given that this is the result of an expert elicitation process (Zommers et al., 2020) there may not be a unique

explanation for the final diagram. In some cases, the detailed information may be unreachable, unless one has a record of the expert elicitation processes beyond what is reported in the final publications. However, when looking at the assessed risks from a distance, it becomes very useful to have a synthetic description of the risks and transitions assessed in each ember. Given the difficulty of describing each of the embers, herein we perform this documentation effort for a selection of embers only, including some that are particularly useful to get a global overview of risk, such as the RFCs. The intent is to show how

useful such information gathering and harmonisation could be, for example for scientists and teachers to get a general understanding of a given climate risk in terms of what is at stake. We hope that this will motivate a more systematic approach in the future, ensuring that synthetic information about the scope of each ember and the explanation for the transitions is collected as part of the elicitation process.

### 2.1.3    Structure of the database

The database is presented in Fig. 2. The data tables shown with a blue frame store the data specific to each ember: the transitions, the description of the risks and risks changes, and ember-specific metadata such as keywords and the range of hazard change over which the assessment was conducted. For example, some embers were assessed over a smaller hazard range than others, in particular when the ember relates to a scenario (such as the shared socio-economic pathway SSP1 (Riahi et al., 2017)) for which high levels of climate change are not expected. The other tables (in grey) provide (meta)data common

to several embers: information about the groups of embers presented in figures, information about figures including the vertical axis of ember diagrams and the full reference to the IPCC report from which a given figure arose.


Database tables and relationships

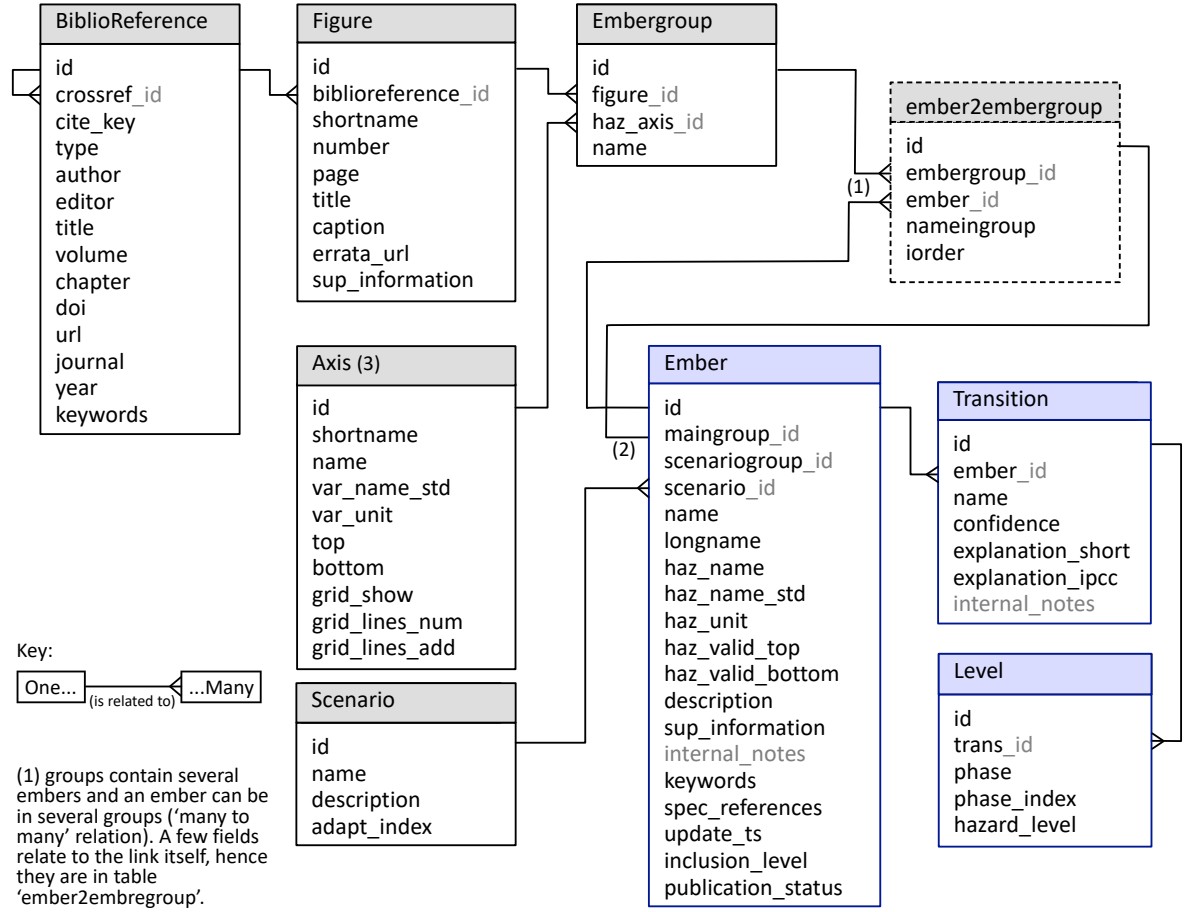

**Figure 2: Database tables, fields and relationships, focusing on the main content (excluding user accounts providing editing access rights and other technical details). Tables with a blue frame store the assessment of risk changes shown in embers diagrams, the other tables provide additional information on how this content should be illustrated and where it appears in IPCC reports. The names prefixed by haz_ relate to "hazard", that is, the variable used as climate change metric (y-axis of the embers). Greyed out fields are editors only (see Sect. 2.5). All variables are described in Supplement S1.1.**

The structure provides the flexibility needed to accommodate all embers and figures configurations existing to date (since the first ember diagram, in AR3) while avoiding error-prone duplication of information. For example, a transition may include complex colour changes with more intermediary levels than the usual median, and an ember may appear in several figures. For full documentation, Supplement S1.1 lists all the fields stored in the database, along with a brief description of their function.



## 2.2 Compiling a homogeneous dataset

Since their first occurrence in the TAR, the design of ember diagrams has changed little, except for the addition of confidence
levels, which first appeared in (O'Neill et al., 2017a). However, numerical values of the assessed change in risk have only
been made available from SR1.5 onwards (IPCC, 2018). Earlier data need to be extracted from the original figures; this was
done for the RFC burning embers in (Zommers et al., 2020; Marbaix, 2020c). Reconstructing the data from the figure
introduces some uncertainty with regard to the risk levels really assessed by the authors, but it is relatively small (about 0.1°C,
Supplementary information in Zommers et al., 2020) as compared to uncertainties in the knowledge of risk levels, and it is
invisible or barely visible on reproduced ember diagrams. Nevertheless, it is evidently useful to systematically store the
assessed risk levels and supporting information at the time of the assessment, as this facilitates understanding, provides
transparency, and makes it possible to verify that the data and the figure are consistent. The last embers for which the numerical
data was still missing were those from the Synthesis report of AR5, which contains the first 'embers' illustrating specific risks
(IPCC, 2014b); their data was extracted herein as previously (Supplement S3, Zommers et al., 2020). The general availability
of information on the embers and their evaluation increased during the AR6 cycle (starting in 2018), and the quantitative data
was also made available by the IPCC through its data portal (IPCC Data distribution centre, 2024 (DDC)), for example in
(Ibrahim Zaiton and Warren, 2023; data from O'Neill et al., 2022). For all embers produced during the AR6 cycle, the data
was obtained directly from the reports, except for chapter 7 of AR6 WGII, for which it is only available from the DDC (Bindoff
et al., 2019; Hoegh-Guldberg et al., 2018; Hurlbert et al., 2019; IPCC, 2022a). The DDC provides a separate file for each
figure of the AR6 cycle containing embers, without ember-specific information outside the numerical values. However,
qualitative and descriptive information about risks and risk levels are critical in order to understand the basis of assessments
or how judgements about risk transitions were made. While getting the numerical data to reproduce the embers has become
easier in the recent IPCC reports, it remains difficult to get a synthetic description of the risks illustrated in each ember and an
explanation for each risk transition. This information is rarely associated with the quantitative data and was not always
collected in a systematic way.

Another important aspect that was not consistently registered together with ember data is a standardised name which uniquely
identifies the climate change metric against which the risk has been assessed. GMST refers to the global mean surface
temperature calculated from air temperature over continents and sea-ice, and sea surface (water) temperature over ocean.
Because model projections provide global mean surface air temperature (GSAT), risks are generally assessed with respect to
past GMST to which GSAT projections are added for the future. In the database, this combination is called Global mean
temperature (GMT henceforth). For a long-time GMST and GSAT had been considered equivalent, but this was challenged
by (Cowtan et al., 2015) who showed models warmed faster in GSAT than GMST. Because of contradictory lines of evidence
from climate models and direct observations, GMST and GSAT are assessed in IPCC AR6 to be approximately equivalent,



with an uncertainty of about 10% (or 0.1°C for present day warming) (see CCB2.3 in (Gulev et al., 2021) for a comprehensive
discussion).

The reference period for GMT increases is also an important piece of information. In AR3, the reference is 1990, with past
changes assessed to be 0.6°C above a period in the second half of the 19th century (figure 19-7 in (Smith et al., 2001) and Sect.
2.2 in (Folland et al., 2001)). In subsequent reports, the period 1850-1900 is used as a proxy for pre-industrial, noting that
anthropogenic changes happened before that period but were small (0.1 [-0.1 – 0.3] °C between around 1750 and 1850-1900)
and more uncertain due to data limitations (AR6 WGI CCB1.2 in Chen et al., 2021). Our database provides GMT changes
above the 19th century baseline of each report; for AR3, this may differ from a 1850-1900 reference by up to ~0.1°C.

Between the 5th and 6th assessment, GMT increased by about 0.19°C (after 2012). In addition, the assessed warming between
1850-1900 and the recent past was increased between AR5 and AR6 on a like-for-like basis owing to incorporation of new
insights about data biases and improved handling of data sparse regions into datasets (CCB2.3 in Gulev et al., 2021). If AR5
GMT levels were updated accordingly, the corresponding impacts would be associated with 0.08°C higher GMT increases
(CCB2.3 in Gulev et al., 2021). However, for the "Reasons for concern" embers, updating the assessment between AR5 and
AR6 led to several changes by roughly 1°C, which were attributed to new evidence and knowledge (IPCC, 2023). Changes to
understanding of the level of warming that would lead to a given impact thus dwarf by an order of magnitude those associated
with new knowledge on long-term warming to date. More generally, re-assessing GMT is only one aspect of the new
knowledge that is obtained over time. Updating earlier embers to AR6 GMT would generate "counterfactual" embers that
would not differ much from the original ones: while the database may help in exploring this change, we refrained from doing
so. Indeed, modifying past embers could be more confusing than useful, particularly as some RFC embers from AR5 and AR6
reports were already copied side-by-side in the Synthesis report of AR6 (figure SPM.4 in IPCC, 2023). In some cases, it might
be difficult to make sure that a scaling of the temperature data is actually the right one as this may require details about the
variables used in the sources of each assessment, which may be either hard or impossible to obtain years after the original
work.

Several IPCC figures involve a conversion of global mean sea-surface temperature (hereafter GMSST) to GMT to illustrate
ocean related risks with the same vertical axis as continental ones. In the SROCC, the IPCC fixed a constant conversion factor
of GMT / GMSST = 1.44 (Bindoff et al., 2019; IPCC, 2019). By contrast, observations over the 20th century suggest values
around 1.25, while conversely, models could be compatible with a scaling factor up to roughly 1.5 depending on the scenario
(Supplement S2, based on (Abram et al., 2019; Fox-Kemper et al., 2021; Gulev et al., 2021; Lee et al., 2021)). Risk estimates
provided in embers are mainly built and interpreted using model projections. Given that the conversion factor derived from





AR6 projections remains quite close to the SROCC value, we keep the same factor (BOX1). This is implicitly done in AR6 WGII, as some of the SROCC embers were reproduced without change (IPCC, 2022d).

Impacts originally occur at a local or regional level: consequently, it is always necessary to 'translate' global metrics such as GMT to changes in local or regional variables which drive impacts, and add uncertainties (cross-chapter box 'CLIMATE' in (Rawshan Ara Begum et al., 2022)). Taking care of this 'scaling' and documenting the approach is thus an important part of the evaluation of how risks increase with global warming, including to produce ember diagrams. There might be difficulties, for example, when an expert elicitation is based on several sources with different approaches, but the communication of the

results benefits from clear indications about the direct risk drivers and how they were linked to the global hazard metric.

**BOX1: From sea-surface temperature to global mean temperature change**

Starting with SR1.5, some embers about impacts on ocean ecosystems and associated services were assessed against global-mean sea-surface temperature (hereafter GMSST), considering that sea temperature is a direct driver of these impacts and that the ocean acidification (decreasing pH) is correlated to GMSST (Hoegh-Guldberg et al., 2018). To illustrate ocean risks side

by side with land related embers, a conversion from GMSST to GMT is thus needed. This was not considered in SR1.5, but the SROCC provided a conversion rule: GMT = 1.44 GMSST (Section 5.2.5 in Bindoff et al., 2019). Is it still a valid rule? Estimating the GMT/GMSST Ratio (hereafter GSR) is challenging due to several uncertainty factors. Without doubt, GSR is larger than one: land areas warm more than ocean, even at equilibrium, a finding supported by available proxy evidence (Eyring et al., 2021; Lee et al., 2021). One of the difficulties is that models consistently project a larger or faster warming of the air

above ocean as compared to surface ocean water, but observational evidence is mixed (CCB2.3 in Gulev et al., 2021). Another issue is that the temperature considered over sea-ice is the temperature of the air, but global warming reduces sea-ice coverage, increasing the ice-free ocean area and complicating the calculation (CCB2.3 in Gulev et al., 2021).

However, instead of looking for "the best estimate" GSR given all information sources, what is needed is consistency with how GMSST was used in the impact analyses. Impacts may depend on local temperature changes, acidification may be an

important driver, etc. For future impacts, the link between specific drivers and GMSST is obtained from model projections (Bindoff et al., 2019; Gattuso et al., 2015): estimating GSR from models only is consistent with this approach. Based on key results presented in the SROCC and AR6, we obtain the estimate of the ratio shown in table 2.





|  | GMT / GMSST ratio (= GSR) |
| --- | --- |
| SROCC (CMIP5, RCP scenarios), projections | 1.43 (1.37 – 1.47) |
| AR6 (CMIP6, SSP scenarios), projections | 1.39 (1.32 – 1.42) |
| AR6, projections + observed changes since 1850-1900 | 1.35 (1.31 – 1.36) |

**Table 2. Estimates of the GMT/GMSST ratio (GSR) based on GMT and GMSST assessed in IPCC reports. For projections, GMT is based on surface-air temperature over ocean as well as over land; results are based on a linear regression across scenarios or time periods (full range between brackets). More information is available in Supplement S2.**


While AR6 values are somewhat lower, the difference with the SROCC is not significant and is inconsequential compared to the uncertainties involved in these estimates or the risk estimates represented in burning embers (the best estimates of the GSR only differ by 3% between reports, as compared to the 4% difference between simulations in the SROCC). Within AR6, the GSR shows no noticeable trend with respect to forcing (supplement S.2), indicating that, given the uncertainties, the use of a

constant GSR remains justified. It is an approximation, especially given that the land-sea warming contrast may change with the scenario in complex ways (Herger et al., 2015; Lee et al., 2021). All values are based on the multi-model mean GMT: we did not use the "constrained" projections based on multiple lines of evidence provided in AR6. Indeed, the objective of these constrained projections is to provide best estimates of GMT for each forcing scenario, without re-assessing the link between mean temperature and other variables, which is our focus here. Interestingly, replacing GMSST by air temperature over ocean

gives a ratio of 1.16 (instead of 1.39) for AR6 projections (supplement S.2): this shows that the difference between air and surface water temperatures plays an important role in these estimates based on model projections (however, as discussed in the main text, there are conflicting lines of evidence on the difference between air warming and water warming, highlighted in (Gulev et al., 2021, CCB2.3)). Future risks assessments would benefit from renewed attention to, and precise documentation of, the specific climate variable(s) which drives the studied impact.

BOX END

## 2.3 Adaptation levels and scenarios

Understanding the extent to which adaptation can affect climate risk levels in the future or according to a given temperature change first emerged as an important component during the AR5 (IPCC, 2014a). While there is consensus on defining adaptation in the IPCC context —i.e. the process of adjustment to actual (in human and natural systems) or expected (in human

systems, or facilitated by human intervention) climate change and its effects, in order to moderate harm or exploit beneficial opportunities (IPCC, 2022c)— there is still no agreed-upon definition of future adaptation levels or adaptation scenarios. As a result, the AR6 assessments included adaptation levels and scenarios using different approaches, either considering the effectiveness of a wide range of adaptation options to reduce climate risks (IPCC, 2022a; Oppenheimer et al., 2019) or deriving an adaptation potential from the SSP framework (Hurlbert et al., 2019). In the former approach, as illustrated in the Europe



Chapter of AR6 WGII (Bednar-Friedl et al., 2022a), authors assessed the effectiveness (low, medium, high) of discrete adaptation options based on the literature and a multi-round collective expert judgement exercise (Muccione et al., 2024), and then assessed how these options could combine together to reach different levels of risk reduction. In the SSP-centred approach, the set of socio-economic pathways explores a range of future societal conditions and related trends in demographics, economics, governance, etc (Andrijevic et al., 2019; Jones and O'Neill, 2016; O'Neill et al., 2017a). These pathways are

constructed to span a range of possible futures with respect to how difficult adaptation —and, separately, mitigation— would be in each socio-economic context. For example, SSP3 and SSP1 respectively challenge or facilitate ambitious adaptation scenarios. As the literature on adaptation frequently used SSPs, the approaches were linked together, for example by assuming that high adaptation happens when there are low challenges to adaptation, such as in SSP1 (even in the Europe Chapter of AR6, some low adaptation embers are based on literature related to SSP1 (Bednar-Friedl et al., 2022a)).

However, making a direct link between 'low challenges' in SSPs and high adaptation is a simplification that may not be entirely obvious. It is consistent because SSP1 includes such hypotheses as effective governance that would make adaptation policies more accessible, facilitating high adaptation, while factors such as slow growth and inequality assumed in SSP3 reduce the adaptation capacity (tables 1 and 2 in O'Neill et al., 2017b). But SSP1 also includes lower population growth and sustainability hypotheses that would inherently result in lower exposure and vulnerability (Byers et al., 2018; O'Neill et al., 2022), reducing

the need for further changes specifically motivated by adaptation. By contrast, scenarios such as SSP3, with hypotheses that generate a baseline with high vulnerability and exposure, could be regarded as needing high adaptation efforts (even more so if the scenario also comes with higher emissions, hence higher climate-related hazards). In summary, SSP3 illustrates a pathway that would result in high adaptation needs but would make high adaptation hard to achieve, while SSP1 illustrates a situation where adaptation is easier but fewer adaptation efforts are required, due to a low vulnerability and exposure baseline.

If high adaptation means achieving low vulnerability and exposure, then it is consistent with SSP1 as commonly assumed; by contrast, if "high adaptation" is defined as "large efforts/changes for adaptation", it could also be justified in the context of a high vulnerability and exposure baseline as defined in other scenarios. Indeed, the SSP framework does not specify adaptation responses: it assumes that these would be defined separately, within "shared reference policy assumptions" (O'Neill et al., 2020). However, developing such common assumptions is challenging, notably because adaptation is highly context and region

dependent (O'Neill et al., 2020). This underlines the difficulty in establishing a common and agreed framework for measuring adaptation levels and benefits, and for designing adaptation scenarios. In this paper, we make a first attempt at highlighting the challenges which we faced while regrouping embers in a common framework, and lay foundations for further explorations as part of the AR7 (see Sect. 4.2).



## 2.4 Overview of the embers compiled in the database

Table 3 provides an overview of the burning embers in the database. Assessments made before the AR6 cycle are shown in light grey. For AR6-cycle embers, the "global" group (green rows) contains all embers from the Special Reports (SROCC, SRCCL and SR1.5) and the two sectoral chapters of AR6 which provided embers (terrestrial ecosystems and health). The WGII contribution to AR6 provides the first "regional" embers, focusing on impacts at the scale of continents (blue lines). Given that impacts have a local nature, this is obviously useful to illustrate impacts in a more insightful way. However, only

about half of AR6 regional chapters provided embers: Africa, Australasia, Europe, North America, along with cross-chapter papers focusing on the Mediterranean and polar regions. Missing continents are Asia and South-America, and there are no embers focusing on, for example, small islands (although there are embers focusing on coastal flood risks).

A total of 19 risks were assessed within a "high adaptation" context. All these embers were also assessed for at least one of the lower adaptation levels, thus illustrating how far adaptation could serve to reduce risks.



| Report: main figure | Short name (title) | Adaptation Low/ med. | Adaptation High | Total | High risk at mean T (min, max) |
|---|---|---|---|---|---|
| AR5-SYR: 2.5 | AR5-SYR CO₂, SLR, rate (Increasing risk from RCP2.6 to RCP8.5) | | | 5 | Not temperature (sea-level, warming rate, CO₂) |
| SR1.5: 3.18 | Coastal and marine life (Risks for specific marine and coastal organisms, ecosystems ...) | | | 9 (+2) [a] | 2.09 (1.15-4.61) |
| SR1.5: 3.20 | Natural and human systems (Risks (...) for specific natural, managed and human systems) | | | 7 (+1) [b] | 1.68 (0.30-3.00) |
| SRCCL: 7.1 | Land overview (Risks to selected land system elements (...)) | | | 7 (+3) [c] | 1.79 (0.90-3.50) |
| SRCCL: 7.2 | SSP (Different socioeconomic pathways affect levels of (...) risks | 3 | 3 | 6 | 1.81 (1.00-3.50) |
| SRCCL: 7.3 | Land-based mitigation (Risks associated with bioenergy crop deployment (...)) | (SSP3) | (SSP1) | 2 | Risk of response measures (no direct link to GMT) |
| SROCC: 5-16 | Ecosystems (Risk scenarios for open ocean and coastal ecosystems (...)) | | | 14 | 3.22 (0.58-5.76) |
| AR6-WGII-2.11 | Terrestrial ecosystems key risks (Key risks to terrestrial and freshwater ecosystems (...)) | | | 5 | 1.69 (0.60-3.00) |
| AR6-WGII: 7.9 | Health (Climate sensitive health outcomes - 3 adaptation scenarios) | 12 | 6 | 18 | 1.67 (0.80-3.00) |
| AR6-WGII: 9.6 | Africa: Key Risks (Key risks for Africa increase with increasing global warming) | | | 3 | 1.50 (1.10-2.00) |
| AR6-WGII: 11.6 | Australasia: Key Risks (Nine key risks for low and moderate adaptation) | low: 9 mod.: 9 | | 18 | 1.47 (0.40-3.00) |
| AR6-WGII: 13.28 | Europe: Key Risks low/med adapt. (Key risks for Europe under low to medium adaptation) [only ecosystems, because others in 13.28 also appear in 13.x] | 2 | | 2 | 2.17 (2.00-2.50) |
| AR6-WGII: 13.29 (a) | Europe: Health (Heat stress, moratlity and morbidity) | 1 | 1 | 2 | 2.52 (1.60-3.80) |
| AR6-WGII: 13.30 (a) | Europe: crops (Risks of losses in crop production) | 1 | 1 | 2 | 1.93 (1.10-3.00) |
| AR6-WGII: 13.31 (a) | Europe: water scarcity (...) risk of water scarcity to people (Key Risk 3) | 2 | 2 | 4 | 2.38 (1.30-4.00) |
| AR6-WGII: 13.32 (a & c) | Europe: flood (...) inland and coastal flooding in Europe (Key Risk 4)) | 2 | 2 | 5 | 2.02 (1.20-3.00) |
| AR6-WGII 14.4 | North America: freshwater (North American freshwater risks) | | | 4 | 2.07 (1.20-3.00) |
| AR6-WGII: 14.9 | North America: Tourism (Relative risks to select tourism activities in North America) | 4 | 4 | 8 | 1.98 (0.80-3.20) |
| AR6-WGII: 14.10 | North America: Economics (Relative risks to economic sectors in North America) | | | 7 | 1.84 (1.00-2.50) |
| AR6-WGII: CCP4.8 | AR6 CCP4 Mediterranean (Key risks in the Mediterranean region) | | | 8 | 1.68 (1.00-2.50) |
| CCP6: CCP6.5 | AR6 CCP6 Polar Regions (Relative risks to select assets in the Polar regions) | | | 11 | 1.94 (0.80-3.00) |
| | **Subtotal without RFCs** Excluding high adaptation cases, there are 128 embers Of these, 121 use an hazard scale that is closely related to GMT | 45 | 19 | 147 (+6) | [a], [b], [c] |
| TAR-SPM: SPM-2 | Reasons for concern (RFCs, pre-AR6). | | | 5 | 3.50 (1.25-6.30) |
| Smith et al. 2009 / AR4 | AR4 included an updated assessment of the « Reasons for concern », but not the burning ember diagram, which was published in (Smith et al. 2009). | | | 5 | 2.25 (0.65-5.00) |
| AR5-WGII: 19-4 | | | | 5 | 2.08 (0.91-4.51) |
| SR1.5: 3.21 | Reasons for concern (AR6 cycle) | | | 5 | 1.61 (1.00-2.50) |
| AR6-WGII 16.15 (b) | | 5 | | 5 | 1.58 (0.70-2.50) |
| | **Total** | 50 | 19 | 172 (+6) | [a], [b], [c] |




**Table 3: Overview of figures containing 'burning embers' in all IPCC Reports. Unless the assessment has changed, each ember is included only once, ignoring repetitions in summaries and across reports of the AR6 cycle, even if the figure is slightly different. Until AR6, all embers aggregated risks in all relevant regions (global scope); AR6 includes the first regional embers, shown in blue. "Reasons for Concern" embers, which aggregate key risks across regions and systems, are shown at the end. All embers published**
**in IPCC reports are included in the database. However, a few embers are not included in the analysis and diagrams presented in this paper. These are indicated by the numbers shown in brackets in the table, with a label referring to the following explanations: [a] SR1.5 Figure 3.18: The 'Mangroves' ember (database id: 15) was reassessed with slightly different numbers as part of SROCC and is included there. For the ember related to 'Open ocean carbon uptake' (id: 21) the information relating to one of the risk transitions appears to be inconsistent.**
**[b] SR1.5 Figure 3.20: For the ember related to the 'ability to achieve sustainable development goals' (id: 34), the information is incomplete (id: refers to the identifier of the ember in the database; it can be used as search criteria to get more information) [c]The SRCCL supplementary material provides data for three additional embers. These are included in the database for completeness, but the embers are not part of any figure in an IPCC report. The risks assessed are: coastal degradation (id: 44), food access (id: 48) and food nutrition (id: 49).**

## 320   2.5   Database access

The content of the database can be accessed in several complementary ways: through a web interface, called the "Climate risks

embers explorer", and through HTTP requests or an archive file, which can provide data to external code (figure 3).

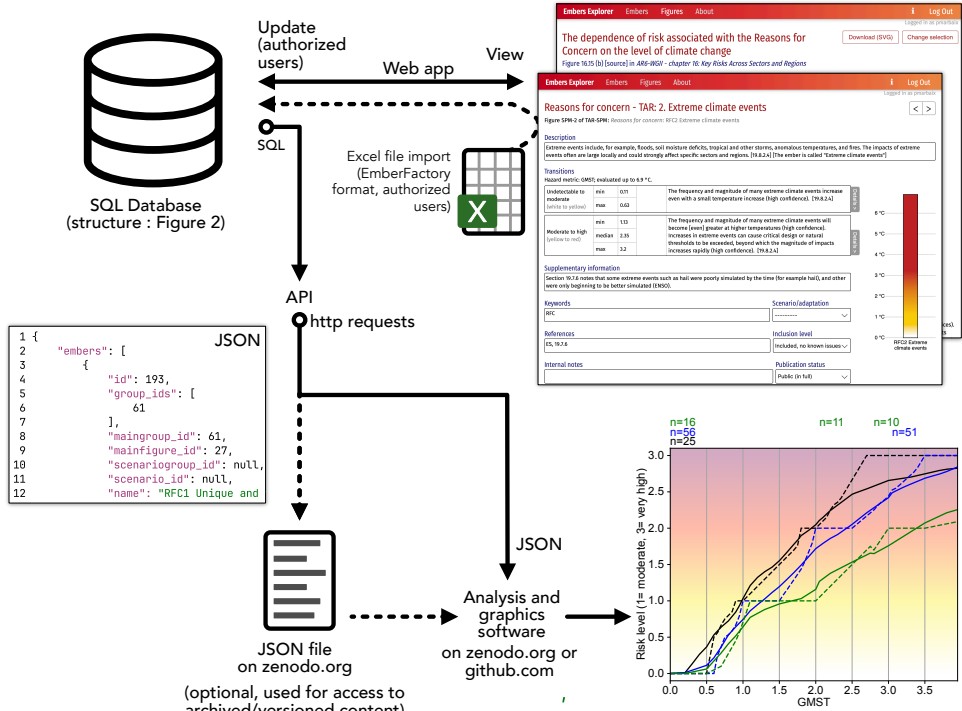

**Figure 3: Schematic view of data access options. Each field in the database and JSON files is described in Supplement S1.**

The numerical data needed to draw the diagram is available for all embers, but the description of embers and transitions is

complete for a fraction of all embers only: filling all descriptive fields is an ideal goal that can only be approached over the





long term. Beyond the publicly available information, the database can store draft text when the 'publication status' of an ember indicates that the description and/or explanation fields are not finalised. This draft information is not used in this paper and is only available to registered editors in preparation for future improvements, as explained in Sect. 2.5.3.

### 2.5.1 Database access from research software


External computer code can access the public content of the database through an Application Program Interface (API). HTTP requests need to be sent to https://climrisk.org/edb/api/combined_data; responses are received in JSON format. Requests may target a subset of the embers according to their unique identifier, keywords, long name, source report, or scenario (Supplement S1.3).


The dataset published with this paper (Sect. 6) is an archive of the result of a request for all (public) information in the database, so this file contains the same data as the result of HTTP requests. The structure of the JSON data is largely identical to the structure of the SQL database (illustrated in Figure 2), with the same field names and minor simplifications (made possible by the fact that the file is not intended to be modified, unlike the database; details are presented in Supplement S1.2).

All figures and tables in this paper can be obtained from both approaches (file and API) using the same software (Sect. 6).

### 2.5.2 Climate risks embers explorer


The web interface provides a searchable list of embers, a list of figures, and the possibility to select embers and get them side-by-side in a figure: https://climrisk.org/cree/list. References to the sources in IPCC reports are provided for each data and related information. The ember diagrams are drawn by the "EmberMaker" software library (Marbaix, 2024b) from the information in the database: it can reconstruct the embers found in IPCC figures, but it does not include additional details, for


which a link to each IPCC publication is provided. The main aim of this interface is to facilitate access to the burning embers and the information needed to understand the risk assessment which they communicate, with a view to being useful to teachers and researchers (Fig. 4).



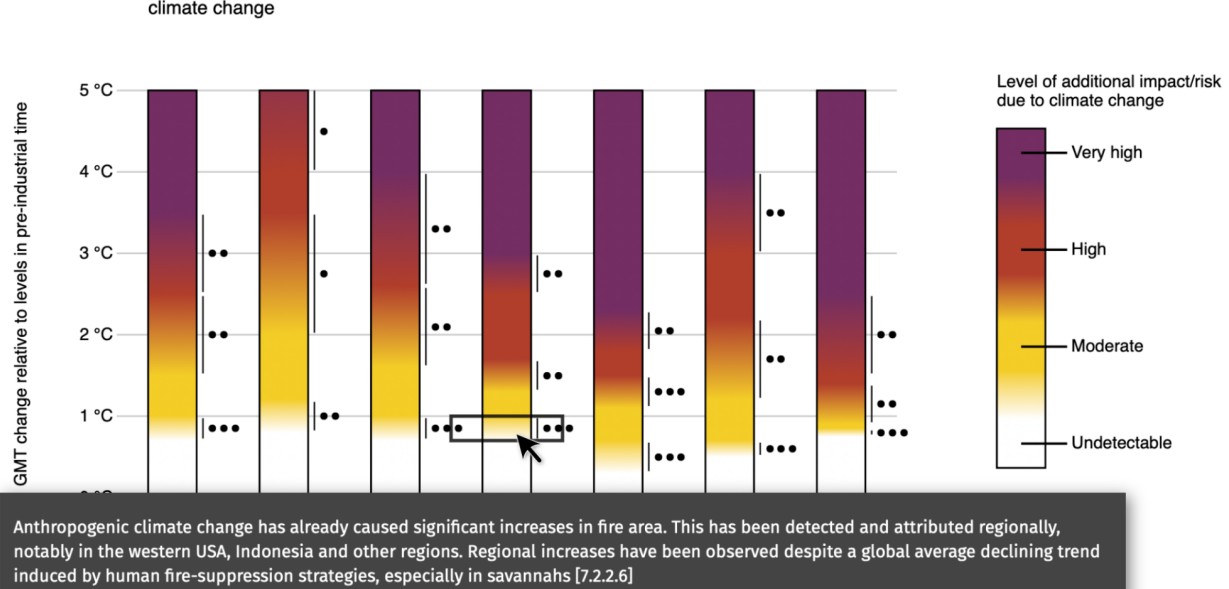

**Figure 4: The Climate risks embers explorer is a web tool which mainly provides a searchable list of embers and a list of related figures in IPCC reports, with full references to these for further information. When descriptive information is available in the database, it can be visualised interactively, appearing on a figure when the user places the cursor over the name of an ember or a risk transition, as shown here.**

### 2.5.3   Enhancing the database through collaboration

As explained above, the current database contains the numerical data for all embers assessed so far and the descriptive

information for a significant fraction of these, but a comprehensive description of all embers is beyond the objectives of this

paper. Improving and supplementing the information about embers assessed in past IPCC reports would benefit from broader

collaborations, notably with researchers involved in each assessment. We have made efforts to facilitate this collaboration by

providing easy ways to contribute. For a quick start, it is possible to download the existing data from the page presenting an

ember (it can be obtained as a Word file) and send an edited copy to the database maintainers. Scientists willing to provide a

larger contribution may create a login and ask for editor status. This identity verification step is necessary to ensure that editors

are qualified and aware of what is expected, and that contributions will be duly recognised. Editing is made as simple as





possible: it is done through the same interface as data consultation. Editors can only change unpublished fields, so that the content subject to editing is not shown to visitors until it is finalised (the publication status is set by database administrators). This process may be adjusted as experience is gained, to make the benefit of contributions available to all while protecting the

database against unwanted changes and ensuring the quality of its content. For this reason, all edits are confidentially logged with the name of the editing user, and care is taken that no content would be lost even in the advent of a technical problem or human mistake (including by sending a full backup of the database to a distant server every day).

As it stands, this database is a contribution to an harmonised documentation of the knowledge synthesised through the construction of burning embers diagrams since 2001, which provides the basis for the analysis in the remaining sections of

this paper. The future availability of this data is ultimately guaranteed by the archive file. It illustrates the potential of structured collection of information to facilitate future assessments as well as to disseminate the results through an interface such as the Climate risks embers explorer. Future development will require continued assessment of the needs and potential benefits, as well as adequate support (see Sect. 4).

## 3 A global picture of impacts and risks

### 3.1 Aggregated measures of risks

#### 3.1.1 Benefits and limitations of aggregated views

While climate change is a global phenomenon, its impacts start from the local interaction of hazard, exposure and vulnerability. However, it is increasingly recognised that impacts are transmitted across systems (cascading impacts) and boundaries (Anisimov and Magnan, 2023; Challinor et al., 2018; O'Neill et al., 2022). As a result of the complexity of impacts, including

the wide range of scales involved, getting an overview of risks is a challenge, and it comes with inherent limitations. The TAR introduced globally aggregated impacts as part of the RFCs, but warned of the limitations of aggregate analysis, which "treat gains for some as cancelling out losses for others" and where the weighting of each impact in the total is "necessarily subjective" (Smith et al., 2001). Authors noted that this masks differences that are important for equity, and added an RFC for "distribution of impacts" to address these differences, in particular the higher vulnerability in developing countries (Zommers

et al., 2020). Aggregating all impacts in a single metric remains challenging, as illustrated by the large range of estimates of global economic impacts presented in AR6 (Cross-Working Group Box ECONOMIC in O'Neill et al., 2022).

As the burning embers are assessed against a qualitative risk-scale (Table 1), aggregation of the estimates across multiple risks requires a mapping of the named levels to a numerical risk index: we use 0 = undetectable, 1 = moderate, 2 = high, 3 = very high risk. This approach was used previously to describe the RFCs (Marbaix, 2020a; Annex IV §56 in UNFCCC, 2015), and

to aggregate the burning embers assessed in the special reports of the AR6 cycle (Magnan et al. 2021). The risk scale can be





seen as similar to the range of colours used in the diagrams, with fractional values representing the colour gradients (transitions) between the named risk levels. However, the necessity to choose a specific index has consequences, in particular when it is a linear one, as calculating a mean risk then means that a high risk is "valued" twice as much as a moderate risk. (Magnan et al. 2021) summarises the limitations of the aggregated risk index as follows: (i) linearity, given that risk could increase faster,

including exponentially and/or with jumps (ii) a limited consideration for systemic feedbacks between different risks, due to knowledge limitations (the aggregation method itself can hardly add such information) (iii) differences in risk valuation among communities and sectors. Incidentally, the burning embers diagrams themselves also have a limitation related to their risk scale, as they cannot reflect the fact that risks may also increase within a given risk level (e.g. within 'high risk', before risks start to meet the criteria for 'very high risk').

Alternatives to the linear scale could be explored, for example if risks are expected to grow quadratically or exponentially as a function of the defined risk levels. The latter would introduce an additional unknown parameter - the growth rate. In the end, whatever choice is made, results will be partly arbitrary. This linearity issue is mitigated when calculating the median risk index among a set of risks, as it only implies that 50% of the assessed risks are larger but not that risks above the median level are 'equivalent' to (or compensate for) those below.

### 3.1.2   How risks increase with warming

Figure 5 illustrates aggregated risks calculated from the regional and global embers developed in the AR6 cycle reports. High adaptation cases are ignored, with a view to obtaining a more homogenous set of data, because there are many risks for which adaptation has not been taken into account. The global picture is that mean risk is increasing by roughly one level (e.g. from moderate to high) for an additional mean warming of 1°C near 1.5 - 2°C; the same conclusion was drawn from AR5 results

almost 10 years ago (UNFCCC, 2015), suggesting that even as we gain more knowledge, some key conclusions remain valid. However, the median risk increases from moderate to high between 1.5 and 2 to 2.3°C, almost twice as fast over that range (see below).

At a given temperature level, both median and mean risk levels are generally higher in the regional chapters than in the global chapters. The difference is very small below 1 - 1.5°C (GMT increase above pre-industrial) and increases to reach about half

of a transition between risk levels around 3°C. In the regional group, very high risk is reached between 3°C and 4°C, while the "global" group stays closer to high risk. There are at least two reasons why we might expect such differences: first, the available embers do not comprehensively cover risks among regions and systems (table 1). We do not know if a more comprehensive geographical coverage would give different results. Second, the risks are taken into account in different ways in the regional and system based analyses; in particular, if a risk is high in several regions, it will 'count' several times in the regional group,

but in the global group it will only appear once, and the aggregated results may show to a lower global risk if some regions



are little affected (at the regional level, a risk might be widespread, which is a criteria for assessing a risk as high or very high, while it may appear less common from a global perspective).

As an illustration of how the focus on certain risks and regions may influence the results, we test an alternative approach intended to bring a more constant weight "per main topic" instead of per ember: a weight is attributed to each ember so that
the total weight is the same for each of the chapters (for AR6) or for each figure (for SRCCL and SR1.5). Those weights are applied to the calculations of mean, median, and other percentiles (figure 5(e)). The results of the 'per ember' and 'per chapter' uniform weightings are very similar. With the per-chapter weighting, the regional and global risk levels are roughly identical up to 1.0°C, then the difference between regional and global moderately increases, with similar levels of risks at all warming levels as in the 'no weighting' case. The large similarity of this result with the previous one tends to support the argument that
the way risks are assessed in regional chapters, as compared to the systems / global chapters, could possibly result from an increased attention, in the regional chapters, to risks that might be high in specific regions, as hypothesised above. The difference between regional and system chapters shown here may then encourage paying attention to the way in which risks are expressed in the global analysis per system affected, when the risks are not geographically homogeneous.

To further illustrate how risks change with temperature, Figure 5(d) shows "aggregated embers" based on the median risk
levels among embers (from Fig. 3(a)). For example, both regional and global aggregated embers show that under about 0.6°C, more than 50% of the risks were considered undetectable. Above that, the majority of risks are increasing, with more than 50% of the embers reaching moderate risk at roughly 1°C (a level significantly exceed by now, with the GMT increase 2014–2023 estimated to 1.20 ± 0.12 °C with respect to 1850-1900 (WMO, 2024)). The transition is smoother for mean risk, which only reaches a moderate level between roughly 1.2 and 1.4°C (approximately where we now stand globally); this is related to the
fact that a few risks do not reach a moderate level until 1.5°C (regional group) or even 2.5°C (global group), as shown by the 10[th] percentile in panel (b). For a typical ember (at the median risk level), the transition from moderate to high occurs between 1.5 and 2°C (regional group) or 1.5 and 2.3°C (global group). Given the definition of high risk, this means that more than 50% of the risks illustrated by embers are expected to become severe and widespread in those temperature ranges. This new illustration of aggregated risks confirms that impacts will escalate with every increment of global warming (IPCC, 2023),
especially above 1.5°C. The year 2023 almost reached 1.5°C already (due to the combination of climate change and climate variability, WMO, 2024), and AR6 concluded that in most scenarios, this level would be reached on a multi-year average in the first half of the 2030's. Even keeping a 66% chance of staying below 2°C would require rapid and deep, in most cases immediate, greenhouse gas emission reductions (IPCC, 2023). Above 2.5°C, 50% of the risks assessed at the regional level start to transition to very high, and are expected to be very high around 3.2°C. This means that more than 50% of assessed
systems would face persistent and/or irreversible adverse impacts, and reach adaptation limits – which have already been reached in some sensitive ecosystems (O'Neill et al., 2022).





So far, we have excluded the RFC embers from the analysis, as they are already providing an aggregated overview of risks. It is thus interesting to compare our results with the two RFC which have the most similar objectives: RFC4, focusing on global aggregate impacts, is an obvious candidate. RFC3, which illustrates the distribution of impacts, may share similarities with our aggregation of regional embers, even though it was not built precisely for the same objective. While our embers and the RFCs should not be expected to be identical, panel(d) confirms that they are quite close. A fraction of this 'close matching' may be due to chance, but the main features of the aggregated embers are robust. This tends to confirm, through a partly independent analysis, that RFC3 and RFC4 do reflect a wide assessment of how risk increases with further global warming and that the RFCs are indeed strongly grounded in the underlying in-depth assessments as would be hoped and expected.








**Figure 5: Aggregated risks based on all burning ember diagrams from the AR6 cycle (AR6, SROCC, SRCCL and SR1.5), excluding the Reasons for concern (shown separately) as well as "high adaptation" alternatives (which were not considered in the RFCs and are included in figure 6). All panels except for panel (c) separate global assessments (focusing on systems) and regional ones (focusing on continents). The total number of included embers is 121 (for more information, see Table 3).**

**Panel (a): solid lines show the average risk level for each global mean temperature increase; dashed lines show the median risk level across embers. The small 'glitches' in the global mean around 2.5, 3, and 3.5°C result from the reduction in the number of available embers because risks were not assessed above a certain temperature level (in particular, beyond 2.5°C in SR1.5); the number of embers taken into account is indicated on top of the figure. The upper limit of GMT is set to 4.0°C because there are even fewer 'embers' assessed above that level (in addition, very high levels of change may result in large uncertainties).**

**Panel (b) shows the 10 and 90th percentiles among the sets of embers, similarly to panel (a).**

**Panel (c) indicates the fraction of assessed embers, at each GMT, for which the risk is above the midpoint within each transition. The left part of panel (d) compares embers constructed from the aggregation of regional or global embers. Transitions are based on the median risk among the set of embers, at each temperature level. The right part of panel (d) reproduces the Reasons for Concern (RFC) #3 and #4 from AR6 (RFCs are excluded from all other parts of this figure).**

**Panel (e) shows the same information as panel (a) except for weighting each ember in a way that allocates the same weight to each chapter or figure (see text)**

**Panel (f) is the equivalent of panel (d) with weighting based on chapters or figures.**

### 3.1.3   The mean and other aggregate metrics: a broader view on risk changes

We have thus far mainly concentrated upon the behaviour of the medians. Other approaches such as looking at the means may

help in getting a broader view on risk changes. The mean and the median can exhibit distinct behaviours, particularly at higher

levels of warming (above 3°C GMT), where the median risk is systematically higher than the mean risk. However, this is

where the mean is less relevant: there is a 'saturation' effect due to the absence of risk levels beyond very high. As shown in

Fig. 5(b), the 90th percentile reaches very high risk at 2°C or 2.5°C. Beyond this temperature level, more than 10% of the risks

are at the top of the risk scale ('very high'), so that the assessed level of risk cannot increase further. This may reflect saturation

effects in the embers: for example, human lives and species cannot be lost twice – that is to say there are certain impacts which

once they eventuate cannot get any worse. Nevertheless, the risks will continue to increase beyond 2.5°C: systems that have

been little affected so far may become severely impacted, with interactions and cascading effects (Sect. 4). Panel (c)

supplements this information by looking at another aggregated metric: it shows the cumulative number of embers for which

risk is beyond the midpoint in a transition. For example, looking at 1.5°C, about a third of the assessed risks are at least halfway

between moderate and high risk, while a little less than 10% of risks are already halfway to very high risk.

### 3.1.4   A closer look at specific risks which may "stand out"

A specific group of embers that may require more attention relates to the risks which remain low even above 2.5°C. The 10[th]

percentile (p10) shown in panel (b) suggests that low risks are more prevalent in the global group: what would be the possible

cause(s), and could it adversely affect the aggregate results? Table 4 lists the specific embers which contribute to the lower

and higher risk percentiles, for a GMT increase of 3°C. A distinctive feature of global embers showing risks up to p10 is that

they all relate to ocean or coastal systems or services. In the case of coastal systems, this could be partly related to a possible





overestimation of the GMT (used for aggregation) as compared to the GMSST that was assessed. This would be the case if, while the reports mention global-mean GMSSTs as the hazard metric, it was in practice difficult to account for the difference between coastal GMSSTs and ocean averages. If the risk assessment relates to coastal temperatures, converting to GMT may

require a lower factor than for open ocean temperatures. This would result in smaller increases in GMT for a given level of risk or, in other words, it would increase the risk at a given temperature. If such a (moderate) underestimation of risk exists, it may possibly contribute to the large number of low risk estimates for coastal risks. However, there are also a number of 'high risk' embers which relate to ocean and coastal systems, as shown in the p90 category in Table 4. All in all, we do not think that this is a game changer for the aggregated results, but it emphasises that checking and harmonising the details of

assessments can be useful. The evolving knowledge, for example related to a potential weakening or collapse of the Atlantic meridional overturning circulation and its consequences for marine ecosystems (e. g. Boot et al., 2024; Van Westen et al., 2024), suggest that updating the risk assessment for ocean systems could be an important focus in AR7, especially given that the related embers were not updated in AR6. Future studies should pay renewed attention to how local sea surface temperature, acidification and sea level relate to GMT (or other global hazard metric), as this was a difficulty in SR1.5 and to some extent,

the SROCC (Sect. 2.2).

Another potential contributor to the higher incidence of low risk in ocean systems is that while most chapter authors have limited their assessment to 'key risks' selected for their severity (Sect. 1), ocean-related embers may include impacts that are not expected to become severe, except possibly under large emission scenarios (e.g. 'vents and seeps' and 'abyssal plains', see Sect. 3.3). Within the regional embers, a large proportion of systems at very high risk is in Australia / New-Zealand (Table 4).

This may reflect the larger occurrence of endemic and/or otherwise particularly vulnerable systems, including corals, although we cannot rule out some heterogeneity in approach between chapters. It also highlights that other regions of similarly very high risk may be missing due to the incomplete coverage of regions in the AR6 embers.




| Global chapters (AR6, SR1.5, SROCC, SRCCL), risks at 3°C GMT (total: 57 embers). | | |
|---|---|---|
| p10 corresponds to **moderate risk**; it includes systems/sectors at: | | p90 corresponds to **very high risk** |
| Close to undetectable risk | Moderate risk | Very high risk includes the following embers |
| Ocean vents and seeps Abyssal plains | Fisheries in mid and high latitudes (fin fish) Eastern boundary upwelling systems Cold water corals Estuaries Mangrove forests Sandy beaches | Warm water corals Marine organisms: pteropods (high- latitude) Bivalves (mid-latitudes, ecosystem impact due to warming and acidification) Fin-fish Terrestrial Ecosystems Terrestrial and freshwater ecosystems: biodiversity loss Food supply instabilities Permafrost degradation Wildfire damage Water scarcity and desertification in drylands Land degradation Food security in scenario SSP3 Ozone-related mortality with limited adaptation |
| Regional chapters (AR6), risks at 3°C GMT (total: 64 embers). | | |
| p10 corresponds to **high risk**; it includes systems/sectors at: | | p90 corresponds to **very high risk** |
| Moderate risk | High risk | Very high risk includes the following embers |
| Antarctic: marine mammals North America: construction | 18 embers, of which 6 relate to coastal or marine systems or services | 30 embers. A few risks assessed with two levels of adaptation are present in this group (above p90): low and moderate adaptation, hence the same risk is included twice (in a different context). When counting these 'duplicates' only once, 25 risks remain, of which 8 relate to coastal or marine systems or services, and 8 are located in Australia or New-Zealand. |

**Table 4: Embers contributing to the lower (p10) and higher (p90) percentiles, at 3°C GMT. Ember names are in italics (for more information on specific risks, see Fig. 7 and 8 or the online tool presented in Sect. 2.5). Given their prominence in the 'global' group, risks related to ocean or coastal systems are highlighted in blue. In the 'global' group, all the risks evaluated as undetectable or moderate come from the SR1.5 or SROCC. The percentiles are chosen for comparability with figure 5; as the risk scale stops at "very high", at 3°C there is a significant "saturation effect" (see text), particularly for the regional chapters, for which the 50th percentile is already close to "very high risk" (Fig.5 (a)). As in Fig. 5, the assessments with "high adaptation" are not included.**

A potentially useful lesson for future reports is that looking at the distribution of assessed risks in terms of severity at given levels of warming may reveal interesting features. This method, or other ways of looking for commonalities between assessed risks, may help to construct a synthesis and possibly identify differences between groups of results that would warrant investigation to distinguish methodological causes (which may suggest further harmonisation) from substantive ones (Sect. 4).




### 3.1.5    Confidence in assessed risk levels

The Ember Database also makes it possible to obtain an overview of the confidence levels attributed to the assessed risk
transitions (for the embers of the AR6 cycle). About 40% of the transitions were given high or very high confidence in both
the regional and global embers (Table 5). Among global embers, 19% were given very high confidence for the transition from
undetectable to moderate risk; conclusive evidence appears to be more readily available for this transition: it largely comes
from past events, and the moderate risk level does not require evidence of widespread risks, unlike for higher risk levels. Very
high confidence is almost never reached for other transitions. At the opposite end of the spectrum, about 15% of the transitions
were assessed with low confidence, with a majority of this for the transition to very high risk. Larger risk and longer term risks
might be commensurately harder to assess and irreducibly more uncertain due to a number of factors. Low confidence is related
to limited evidence, which may be due to the lower availability of studies about large changes, to which a focus on 1.5°C and
2°C following the Paris agreement may have contributed (Kemp et al., 2022). Reaching firm conclusions might be more
difficult because risks result from a combination of more uncertain factors, related to long-term societal changes and/or
regional climate projections. However, the link between uncertainty and confidence is complex, because uncertainty can be
represented by a wide transition range, which tends to increase the confidence that the transition is indeed within that range
(Zommers et al., 2020); this is illustrated in the assessment of the RFC related to aggregate impacts in AR6: two ranges were
assessed for the same transition, with more confidence in the larger one (O'Neill et al., 2022).




| Global chapters (AR6, SR1.5, SROCC, SRCCL) | | | | | | | |
|---|---|---|---|---|---|---|---|
| Transition | Mean GMT (°C) | Mean confidence (index) | **Number of embers** for which this transition was assessed with the confidence level | | | | Total number of embers |
| | | | Low | Medium | High | Very high | |
| undetectable to moderate | 1.1 (recent past) | High (2.8) | 4 | 13 | 29 | 11 | 57 |
| moderate to high | 2.2 | Medium (2.2) | 7 | 33 | 16 | 1 | 57 |
| high to very high | 3.1 | Medium (2) | 10 | 22 | 6 | 1 | 39 |
| Fraction of all transitions (for all embers) | | | 14% | 44% | 33% | 8% | |
| Regional chapters (AR6) | | | | | | | |
| Transition | Mean GMT (°C) | Mean confidence | Low | Medium | High | Very high | Total |
| undetectable to moderate | 0.8 | Medium to high (2.5) | 4 | 23 | 37 | 0 | 64 |
| moderate to high | 1.7 | Medium to high (2.3) | 7 | 30 | 25 | 0 | 62 |
| high to very high | 2.8 | Medium (2) | 16 | 31 | 12 | 1 | 60 |
| Fraction of all transitions (for all embers) | | | 15% | 45% | 40% | 0.5% | |
| "Mean confidence" is the mean of a confidence index defined as 1 = low confidence, 2 = medium confidence, 3 = high confidence, 4 = very-high confidence. Mean GMT and confidence are rounded to the first decimal place. | | | | | | | |

**Table 5: Confidence levels for the risk transitions. The central part of the table provides the number of embers which received a given confidence level for a given transition. The second column indicates the mean GMT of the median of each transition (e.g. halfway between undetectable and moderate risk). The last column is the total number of embers for which a given transition was**
**assessed.**



### 3.2 Human vs natural systems, and the potential role of adaptation

Figure 6 aggregates embers in 3 groups: risks for ecosystems, risks for other systems (including some ecosystems services) excluding the assessments considering high adaptation, and remaining risks with high adaptation. The 3 groups are exclusive, as none of the embers about ecosystems consider a 'high adaptation' case. Results show that aggregated risks are generally higher for ecosystems compared to human systems, consistent with (Magnan et al., 2021). Part of this result may be due to the limitations of autonomous adaptation in natural systems (especially at large warming rates), as well as insufficient knowledge regarding the effectiveness of different human interventions to support natural adaptation. As a result, authors of the AR6 chapter on Europe, for example, did not produce embers on terrestrial or marine ecosystems under high adaptation (Muccione et al., 2024). Another potential explanation for higher risks in natural systems is that these involve a more "direct" connection between climate and risks as compared to some of the human systems, which also depend on less- or non- climate-sensitive factors (e.g. urban planning, part of the economic production, etc.).

For risks affecting human systems and ecosystem services, the potential benefit from adaptation appears significant, for example reducing the median risk at 2°C from high to moderate. Up to 2°C and under high adaptation, more than 50% of the risks studied are still considered moderate. The same applies without high adaptation up to just over 1.5°C. For the studied set of risks, adaptation helps gaining 0.5°C for the start of the transition to high risk, and 1°C for the point at which more than 50% of the risks are assessed as high (end of the transition to high risk).



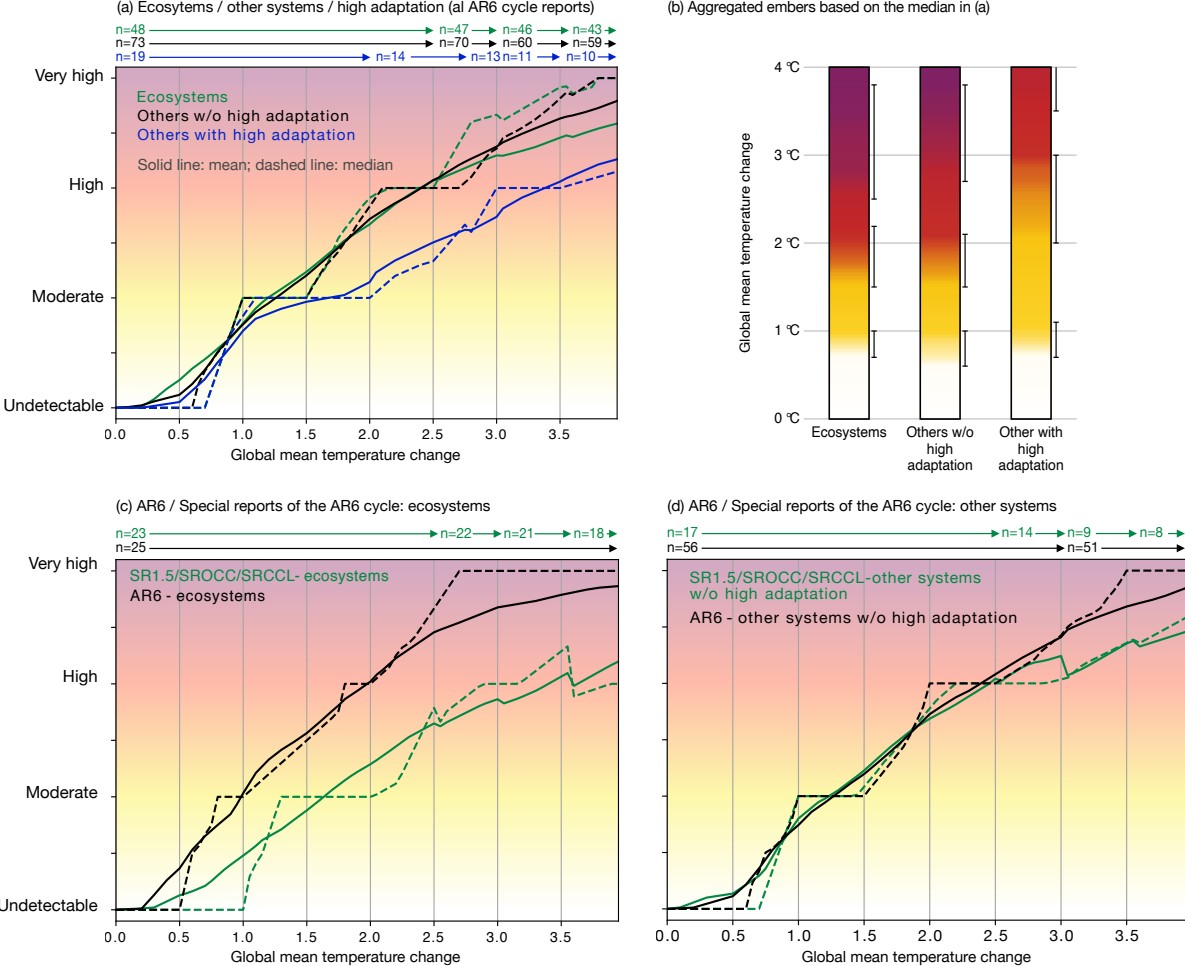

**Figure 6: Aggregated risk levels for ecosystems (panel (a): green), all other risks assessed with a level of adaptation that is either
undefined or at most moderate (black), and embers with a hypothesis of high adaptation (blue). Solid lines denote mean values and
dashed lines show the median among embers. The number of included embers is provided on top of the left panel.
Panel (b): representation of the median values as burning embers (note that the volume of available data decreases as warming
increases (see text), as shown in panel (a); the embers reflect the available data at each level of warming, in particular for the
adaptation cases: there is no implication that adaptation would be "feasible and efficient" for all systems at all levels of warming).
Panel (c): separation of the risks for ecosystems shown in (a) by source – AR6 or Special reports.
Panel (d) same separation, for other systems (without high adaptation).**

Some of the "high adaptation" cases were not assessed above 2°C. This is notably the case when the analysis is based on a

scenario that assumes socio-economic conditions favouring ambitious mitigation (in particular SSP1), thus avoiding higher

temperature increases. The few cases which were explicitly assessed in relation to scenarios involve scenarios SSP1, SSP2 and

SSP3, which form a subset of the SSPs in which challenges to adaptation and mitigation are supposed to increase or decrease

together (O'Neill et al., 2017b). As the assessment does not cover temperatures beyond those projected in each scenario, the



corresponding embers were truncated to the nearest whole °C of projected warming in 2100 (IPCC, 2022d, 2023). Consequently, fewer 'high adaptation' scenarios are available at higher warming levels in the aggregated analysis. This situation may partly result from the selection of SSPs that was included in the available studies, as the SSP framework itself would in principle allow other combinations of mitigation and adaptation (especially within SSP5 and intermediary scenarios that would still involve substantial use of fossil fuels). The alternative would be that the combination of high warming and high adaptation is less plausible, if aspects of sustainability, as idealised in SSP1, are more effective in reducing risks when progressing together in a broader context. There is support for this hypothesis in the WGII contribution to AR6, as it concludes (with medium confidence) that prospects for climate resilient development – including mitigation and adaptation – will not be possible in some regions and subregions if the global warming level exceeds 2°C (IPCC, 2022d). This relates to the issue of adaptation limits, for which important knowledge gaps remain (Berrang-Ford et al., 2021).

### 3.3 Risks across systems: contributing to the identification of hotspots?

To provide a concrete view of the risk changes, Figs. 7 and 8 present data from all embers of the AR6 cycle in a synthetic way, focusing on risks at 1.5°C, 2°C and 2.5°C GMT. This illustrates risks that can be avoided at lower levels of GMT (smaller circles) and via adaptation (lines connecting circles for the same GMT).

### 3.3.1 Risks increase with temperature

Consistent with the aggregated results in the previous sections, each 1°C rise in GMT increases many risks by roughly one category (e.g. from medium to high), but there is considerable variation between risks and risk areas. For example, risks for ocean ecosystems and species at 2°C span the whole range from undetectable (cold water corals, abyssal plains, vents and seeps) to very high risks (warm water corals) – a broader view on risks that is made possible by the large number of embers assessments in the SROCC. The diversity of risk levels is a reminder that it is important to assess and illustrate risks with different characteristics separately via distinct embers, with particular attention to rapidly increasing risks. It is a key justification for the development of the 'non aggregated' Figs. 7 and 8, which supplement the aggregated views.

Comparing risk levels should not give the impression that some risks are identified as more worrying than others, as the nature of risks differ, and the relative importance of risks depends on value judgements (Sect. 3.1 and Smith et al., 2001). With this in mind, it is possible to identify systems that are roughly at high risk at 1.5°C GMT (in the absence of explicit consideration of increased adaptation): food supply instabilities, desertification associated with water scarcity in drylands, coastal flooding and warm water corals. Focusing on regional analyses, high risks at 1.5°C are mainly found for the polar regions, especially the Arctic (sea birds, sea-ice ecosystems, permafrost), Australia and New-Zealand (including coral reefs, other high biodiversity ocean ecosystems, and human settlements), and the Mediterranean region (marine ecosystems and delayed risks of coastal flooding). However this does not mean that risks in other regions remain low, especially in the low latitudes, given



the absence of embers in several chapters, and the limited number of risks assessed in this form for Africa – most of these regions are highly vulnerable and already experienced large impacts (Sect. A.2.2 in IPCC, 2023). It will be important to ensure comprehensive coverage in future reports. To achieve this, the data gap for little-studied regions, particularly in developing

countries, needs to be filled - a key challenge for scientific research.

Some risks increase by significantly more than one risk level in the 1.5-2.5°C range, suggesting that limiting warming could be particularly effective to minimise these risks; these include terrestrial and freshwater ecosystems and biodiversity, as well as to some marine systems (bivalves and fish) and land degradation. Regional analyses add other large increases in risks within this temperature range for health (through heat-related mortality and morbidity, and infectious diseases in Africa), coastal flood

risks and food production.

### 3.3.2    Insights on adaptation

In line with table 3, Figs. 7 and 8 show that only a limited number of burning embers were assessed for different adaptation scenarios, a situation that will need to be substantially advanced in future IPCC assessments.

These figures provide a disaggregated perspective on the substantial shift in risk reduction that can be expected from medium

to high adaptation efforts, especially when it comes to avoiding high and very high risk levels. This can be seen for health, food insecurity and desertification (Fig. 7) as well as for the Europe set of embers and some aspects of tourism in North-America (Fig. 8). Australia-NZ sets exhibit shorter lines joining risk levels without and with adaptation, but high adaptation was not considered and the result still shows room for manoeuvre in terms of risk reduction. Summing up the cases for which adaptation has been considered, the risk reduction potential appears to be substantial, at least up to a warming level of 2.5°C.

However, high levels of adaptation may require transformative changes that may involve trade-offs, and risk reduction through adaptation will not be equally effective in all sectors nor in all regions and social groups. Such conclusions will have to be further explored in the future as there is still a pressing need to better harmonise the way adaptation scenarios are included in risk assessments across sectors and regions (see Sect. 4.1).

A few examples in the database illustrate that introducing different adaptation scenarios in risk assessment may allow for a

more in-depth assessment of the plausible range of future risks. For example, SR1.5 assesses heat-related morbidity and mortality considering only autonomous adaptation, and concludes on a risk range from above moderate to high (for the illustrated GMT range: 1.5 to 2.5°C); the inclusion of three adaptation scenarios however allows the AR6 WGII to identify, for the same health risks and temperature range, a wider risk range, from moderate to very high (this range thus extends both under and above the previous estimate, depending on the adaptation level).





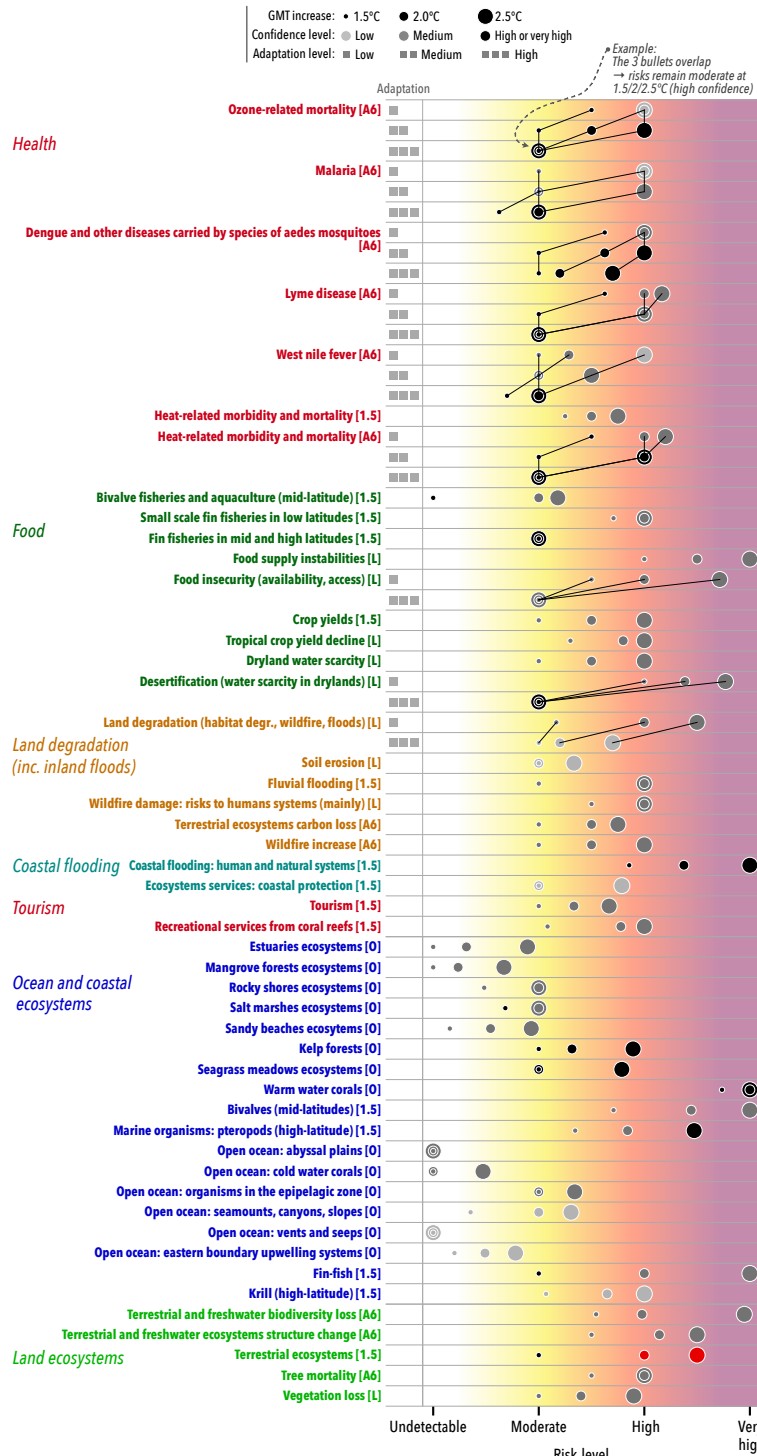

**Figure 7: Risks to different systems, as assessed in embers presented in special reports (SR1.5 – noted [1.5], SRCCL [L] and SROCC [O]) and in AR6 (A6). Three levels of warming (1.5, 2 and 2.5°C GMT) are represented by increasing dot symbol sizes. The thin lines connecting the dots represent potential risk reduction through adaptation (square symbols on the left indicate adaptation levels). GMT increases beyond 2.5°C are not shown because they are not available for some embers (from SR1.5 or with high adaptation).**




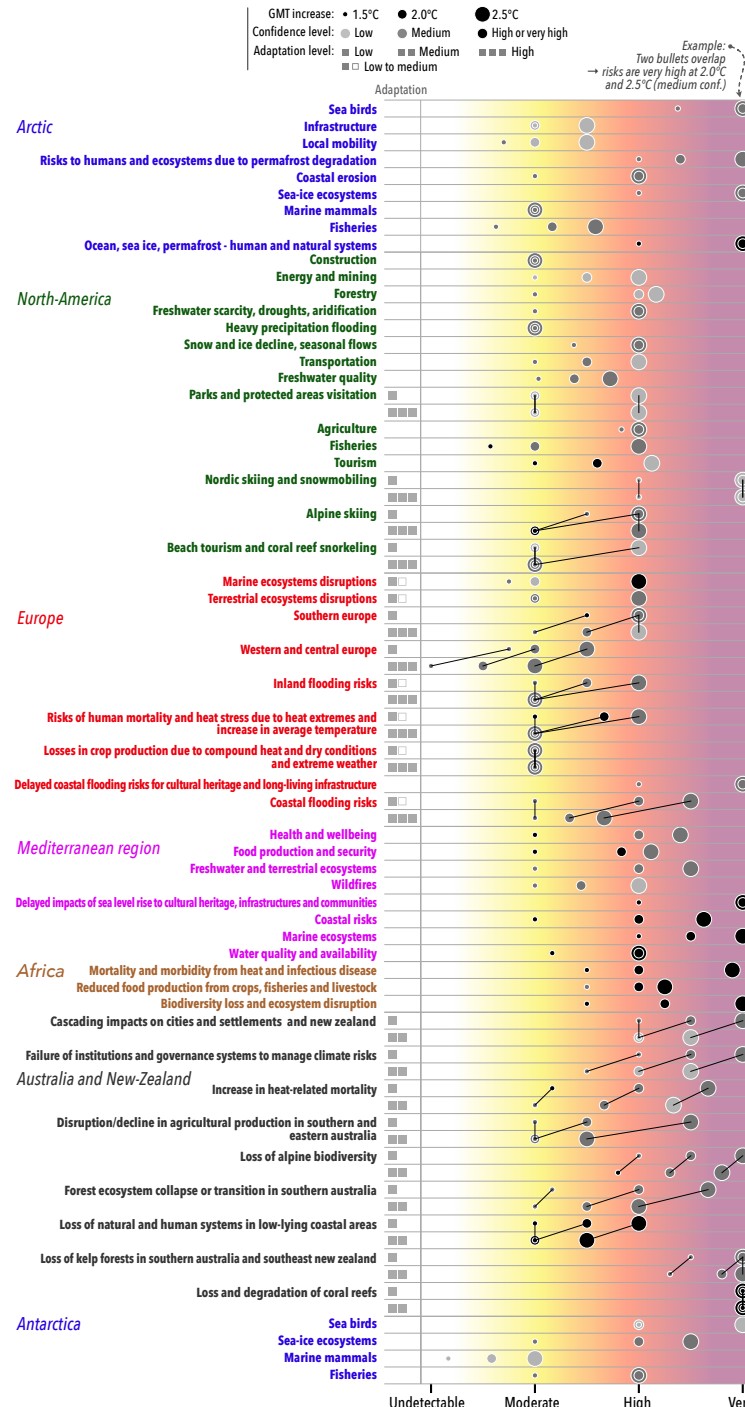

**Figure 8 – Risks to different geographical areas, as assessed in embers from AR6 regional chapters, and in two embers devoted to risks in the Arctic (as indicated by the sources, SR1.5 [1.5] and SRCCL [L]). The symbols have the same meaning as in Fig. 7.**



## 4 Discussion: contribution to future risk assessments and communication

### 4.1 Towards a more comprehensive overview of climate risks

Burning embers can represent a summary of climate change risks in specific systems and regions in a consistent and usable form. Aggregating these assessments gives an overall picture of risks that is consistent with what has been concluded from other approaches: notably, half of the assessed risks have already reached a moderate level. If global average warming reaches around 2°C, half of the risks assessed at a regional level will reach a high level, meaning that region-specific risks will be severe and widespread within a given region. If a high level of adaptation is achieved in every human system or activity, which would involve challenges and require profound changes, risks could remain moderate up to 2°C in around half of the assessed cases – a substantial risk reduction, although one should not ignore the fraction of risks that would nonetheless become severe. This remains an early result with several limitations due to a limited scope - for example, the potential for reducing risks for ecosystems is little covered, while any such reduction may be harder to achieve than for human systems. Differences in adaptation level were considered in only about 20% of the risks illustrated by embers to date. As it provides a more comprehensive view of how risks may evolve, future reports would benefit from a systematic consideration of the impacts of adaptation actions in a similar way, and possibly from further investigation on how differences in exposure and vulnerability could be included (O'Neill et al., 2017a). Our synthesis also shows that the combination of high warming and high adaptation has only been considered for about half of the high adaptation cases, with the other cases assessed only up to 2 or 3°C of global warming. As this is due to the characteristics of the socio-economic scenarios considered in the underlying studies, the need to assess adaptation and its limits at higher levels of warming should also be taken into account in future research. Another limitation of the overview provided by the compilation of embers is that it only takes into account the interactions between drivers and/or risks, including compound and possibly cascading risks, that were considered in the original assessment: at each level of warming, impacts may be amplified by interactions within or across regions and systems presented in different embers (Berrang-Ford et al., 2021; Simpson et al., 2023). Regions and risks not considered in the underlying assessments may also significantly shift results compared to those sampled within AR6 assessments.

Looking at individual embers gives insights into the systems that AR6 authors assessed to be most affected: human and natural systems in coastal areas are among these. Food and water availability could be compromised at least where a high level of adaptation will not, or cannot, be achieved. At 2°C GMT increase, several types of ecosystems of great importance for biodiversity are among those at high risk on land, in coastal waters, and particularly in the Arctic. However, the significance of this conclusion is limited by the absence of burning embers for some systems or sectors and about half of the regions, which does not mean that these are less affected. The information needed was often missing or incomplete, such as in the AR6 WGII





cross chapter paper on mountains, which assessed key risks and could estimate the risk levels for some of them, although it did not consider the available evidence sufficient to build embers (Adler et al., 2022). The absence of embers may have other causes, such as for example in AR6 WGII chapter on Central and South America, which produced a diagram for the second order draft (SOD) but deleted it for the final version, with responses to review comments stating that this was due to difficulties

in implementing the expert elicitation methodology (Castellanos et al., 2022; figure 12.7, SOD chapter 12 in IPCC AR6 drafts and Review Materials, 2024).

How can future burning ember illustrations help to further clarify the global picture of risk? First, we need to aim for the broadest possible coverage of regions and systems at risk. This should include systematically conducting a risk assessment for all regions. It would update and complement a similar summary in AR5, which could provide a synthesis of key risks for each

region and present the results graphically for two levels of warming and adaptation (Assessment Box SPM.2 in IPCC, 2014c). Further, at the regional level, it is important to separate risk assessments for those parts of a regions where they differ. This was done, for example, for water scarcity in the chapter of AR6 dedicated to Europe, as water scarcity differed between southern and central european regions (Bednar-Friedl et al., 2022a).

Future assessments may wish to use embers to illustrate compound risks and risks associated with adaptation and mitigation

response measures. The SRCCL provides the only example of the latter to date, with embers synthesising the risks to food systems, terrestrial ecosystem and water security associated with the potential increase in land area used for the deployment of bioenergy crops in 2050 to meet mitigation targets (Hurlbert et al., 2019). Two contrasting scenarios were explored. In both, the hazard metric was the amount of land used for dedicated bioenergy crops. Going forward it may be useful to explore ways of defining a framework or common methodology to enable a consistent assessment and synthesis of risks from response

measures. Finally, embers could potentially illustrate the assessed risks associated with temperature overshoot pathways, where global warming more or less strongly exceeds a long-term limit, in particular 1.5°C, and then declines more or less rapidly (Meyer et al., 2022; Reisinger and Geden, 2023).

To facilitate synthesis work and make it even more instructive, one could ask experts to choose, for each ember established in the future, a list of keywords corresponding to the main risk factors. Their main role would be to help in classifying risks

assessed within embers, highlighting the systems and regions most at risk and the common causes of risks increases. Potential keywords may relate to 'regional specificities' such as polar amplification, the appearance of new climatic conditions that have no equivalent in the recent past in tropical regions, and mountain cryosphere – which is rapidly declining with substantial impacts on societies and ecosystems (Constable et al., 2022; Hock et al., 2019; Jia et al., 2019). Other keywords may highlight compound events which play an important role in impacts, such as drought and extreme heat or warming and acidification of

coastal waters, and the presence of socio-economic factors that increase the vulnerability or exposure of populations (Simpson



et al., 2023). To maximise their usefulness, these keywords could be defined iteratively over the course of each project or report. .

Burning embers allow an evaluation of how the assessment of a risk has changed over time when the same risk has been assessed in different IPCC cycles. So far, this has been the case only for the RFCs, as these are the only embers to have been

assessed several times over successive assessments (Zommers et al., 2020). The Synthesis Report of AR6 has already shown that climate-related risks synthesised in the RFCs are assessed at a higher level (or put another way to eventuate at lower levels of GMT) in AR6 than in AR5, due to improved scientific understanding (IPCC, 2023). As many risks have now been illustrated with embers, we have a solid basis to which new knowledge can be added, highlighting risks that are reassessed at a different level and findings that are confirmed, possibly with a higher level of confidence. To realise this potential, we need to maximise

the compatibility of the new results: including risks already considered previously, defining them in the same way, and using the same methodologies consistently, taking care to reinforce them but introducing changes only when necessary, in a well-documented way. The database will then make it easier to track changes over time, both in terms of risk levels and associated textual explanations, by linking the 'embers' for the same risk.

## 4.2    Further exploring adaptation potential to reduce climate risks

The conclusions of this first exploration of the Ember Database on the expected benefits of adaptation in terms of risk reduction align with previous findings from (Magnan et al., 2021). However, as already noted in the earlier study, these results need to be taken with caution because the information developed under the IPCC cycles shows some heterogeneity, especially in terms of the scales and regions considered and the adaptation scenario framing. Future assessments would benefit from a framework that defines metrics for measuring vulnerability and exposure more consistently within and across Working Group II

contribution to reports (including the upcoming Special Report on cities), continuing efforts to consider a variety of possible socio-economic futures as well as a range of adaptation scenarios (Zommers et al. 2020). The IPCC recently announced that it will issue updated technical guidelines on impacts and adaptation in parallel with the WG2's contribution to AR7, considering adaptation indicators, metrics and methodologies; this development can provide opportunities to further integrate adaptation within burning ember diagrams (IPCC, 2024).

In that view, one key question relates to how to define what "ambitious" adaptation means across systems and inherent context-specificities, as well as what adaptation levels being in-between low and ambitious adaptation refer to. While "low" adaptation could be easily defined in relation to business-as-usual practices favouring incremental responses, can "ambitious" adaptation be defined through more transformational practices changing the fundamental attributes of a given system and therefore addressing the root causes of systems' exposure and vulnerability? It is worth exploring such an hypothesis because the

narratives ranging from business-as-usual scenarios to more transformational responses is gaining traction in the literature



(Rawshan Ara Begum et al., 2022) as well as in the policy arena. This is important because major changes motivated by adaptation, particularly of a transformative nature, raise a number of questions such as: what is being transformed and how (societal aspects, including values and norms, livelihoods including for example resettlement, techniques possibly including some that move from nature conservation to actively 'transform' living species and ecosystems…)? To what extent is it

possible to address the risks associated with the extensive transformational adaptation that would be required in high emission scenarios? What about the ethical and justice aspects of transformations imposed by the need to avoid intolerable risks? How does transformational adaptation link to wider socio-economic changes and transformation aimed at ensuring sustainable development, in particular within the concept of climate resilient development pathways (Schipper et al., 2022)?

A potential starting point is the framing developed in one of the synthesis chapters of the AR6 WGII (O'Neill et al., 2022),

which describes adaptation levels based on 4 main characteristics of adaptation-related responses: depth (do the responses support major shifts from the business-as-usual situation when required?); scope (is adaptation widespread, both geographically and across systems?); speed (to what extent is adaptation fast enough to keep pace with increasing and accelerating climate risks?); and limits (do the responses overcome limits?). Table 6 is a first attempt at designing such a proposal. Whatever the final framing, it is evident that further structuring the way adaptation scenarios are considered, linked

to socio-economic pathways, and included in risk assessments will help to assess adaptation benefits in a more consistent way, and therefore ensure more robust aggregated analyses and would likely lead to major progress from the AR6 to the AR7 Summary for Policy Makers and Synthesis Reports.





| Low adaptation | **Depth**: Adaptation action is largely an extension of existing practices at the site/context; it does not challenge underlying values, assumptions and norms.<br>**Scope**: System-level adaptation actions are mostly of a reactive nature (in response to extreme events occurring), are localised and fragmented at the system level (i.e. terms of geographic or system coverage, with adaptation being medium in some regions or systems and low in others), and uncoordinated among the various stakeholders.<br>**Speed**: At the system level, adaptation action remains slow and does not help keeping pace with increasing climate risks.<br>**Limits**: As a result, adaptation does not allow to substantively challenge soft limits. |
|---|---|
| Medium adaptation (here reflecting a continuum of scenario options between low and high) | **Depth**: Adaptation action reflects a shift away from existing practices at the site/context, norms or structures to some extent.<br>**Scope**: Adaptation affects wider geographical areas, multiple areas and sectors within the system, or are mainstreamed and coordinated across multiple dimensions.<br>**Speed**: Adaptation is implemented moderately quickly.<br>**Limits**: Medium adaptation defines an incomplete use of the potential. It helps address some soft limits but does not challenge hard limits. |
| High adaptation | **Depth**: Adaptation action reflects entirely new practices (to the given context) involving deep structural reform, complete change in mindset, major shifts in perceptions or values, and changing institutional or behavioural norms.<br>**Scope**: Adaptation is widespread and substantial, including most possible sectors, levels of governance, and actors.<br>**Speed**: Change is considered rapid for a given context.<br>**Limits**: Adaptation allows to exceed many (if not all) soft limits as well as to substantively challenge hard limits. |

**Table 6: tentative description of clusters of adaptation scenarios based on a 4-fold framing distinguishing between the depth, scope, speed and limit of adaptation-related responses. Adaptation scenarios are to be applied at the study system level, i.e. a sector or a territory, for example. Inspired from (O'Neill et al., 2022).**

### 4.2.1 Adaptation limits and residual risks

Together, the findings in sections 3.2 and 3.3 and the above discussion on the further use of scenarios that include adaptation raise the question of how to capture the limits to adaptation. That is, at what level of further climate change do we move from the level of risks that can be reduced through high adaptation, to risks that will remain even after ambitious adaptation action. Limits to adaptation may depend on trade-offs with other objectives, especially at high levels of warming where more extensive planned transformation would be needed to limit intolerable risks and/or undesirable forced transformation (New et al., 2022; Rawshan Ara Begum et al., 2022; Schipper et al., 2022). The fact is that scientific knowledge on adaptation limits and the associated levels of residual risk remains underdeveloped (Berkhout and Dow, 2023). There would be high-value for future assessments and in particular AR7 to further include such reflections and elements in the process of building burning embers and be able to come up with overview conclusions for regions, sectors and globally. That could be ground-breaking information to feed the Loss & Damage mechanism established under the UNFCCC (Otto and Fabian, 2023; UNFCCC, 2014).



## 4.3    Expanding the database

The web interface allows registered experts to edit the descriptive information provided for each ember (Sect. 2.5). Contributions, be they broad or specific, will obviously continue to be welcome, as they would further enhance the usefulness
of the database.  As the information was not always collected at the time of expert elicitation in the past, we do not expect that complete and detailed information will ever be provided for each ember, but we have built up an already useful base and the tools to further develop this documentation.

Importantly, the database can incorporate new assessments. A pragmatic approach is to first prepare Excel spreadsheets in the format which was previously used to draw AR6 embers, and supported by the EmberFactory application (Marbaix, 2020b).
These files can be imported in the database. The same approach could be followed for new embers assessed outside IPCC reports - as long as the data is available as open content, and possibly restricting to peer-reviewed assessments. These might be handled in a way that makes clear their distinction from embers produced under IPCC auspices. As highlighted above, future assessments would greatly benefit from the collection, as part of the expert elicitation process, of several elements beyond the risk levels: descriptive information on the risk and transitions, keywords including information about risk factors,
and link to scenarios setting the vulnerability, exposure, and adaptation context. We hope that the database presented here will help and motivate a more systematic approach to the collection of information.

This, in turn, may motivate further work to improve the database, and to reflect on how online tools can facilitate the expert elicitation process: such tools may reduce the burden of data manipulation during elicitation cycles, improve the feedback provided to the expert team as part of the process, and reduce the risk of error. This could build on the draft tool implementing
part of the process for aggregating views within a group of experts described in (Zommers et al., 2020), which is available at climrisk.org/emberelicitation and could be further developed and integrated within the structure and web interface of the database presented in this paper (climrisk.org/cree). There is substantial potential for expanding the support to each stage of the expert elicitation process involved in the preparation of embers, and to make further use of its outcome. This would likely have the notable benefit of increasing the homogeneity of the assessment undertaken across diverse reports and chapter teams,
adding considerable rigour to the embers as an assessment and illustration tool.

## 4.4    Further use for analysis and communication

While the IPCC already synthesises the information on the risks presented in ember diagrams, for example in the WGII Atlas and the Synthesis Report (IPCC, 2022b, 2023), a more interactive presentation, including the description of risks and transitions presented in this paper, may facilitate access and broader uptake and usage. Concrete examples are available in the

CREE web interface, including for the AR6 Reasons for Concern: climrisk.org/cree/emberfigure?figure=14. The increased accessibility to the data, including the descriptive information may also facilitate the creation of new visualisations.

### 4.4.1 The future of "burning embers"

A key message that comes out, once again, from this overview of risks, is that available knowledge is more than sufficient to motivate urgent action towards stabilising temperatures. As risks are increasing, we would like to express the wish that
"burning embers", which were built to illustrate how risk may increase with warming, will soon become unnecessary in their current form. As the success of mitigation action becomes clearer, the focus of risk assessment may move to a better understanding of regional hazards and adaptation options at the expected level of global climate change, taking into account that stabilising global mean temperature will not halt other changes such as sea-level rise and the melting of glaciers. However, at present, future warming remains uncertain, as there remains a gap between stated global ambitions and current (and planned)
national actions (UNEP, 2023), although the rate of global greenhouse gas emission increase went down over the last decade (Friedlingstein et al., 2023), fuelling hope of an imminent peak (Fyson et al., 2023).

## 5 Data and code availability

The data collected for this paper are available from Zenodo at https://doi.org/10.5281/zenodo.12626977 (Marbaix et al., 2024). The file is in JSON format, which is accessible from many software environments and is text-based. It contains a metadata
section with general information such as when the data was extracted from the database, then a description of each data field. In recognition of the fact that these data are based on the assessment provided in IPCC reports, we ask users of the dataset to include references to the relevant IPCC reports in their publications. These references are provided in the Zenodo record (as well as in the file itself). The data are made available under the Creative Commons BY 4.0 licence.

The code used to generate the figures and tables in this paper is available from Zenodo at https://doi.org/10.5281/zenodo.12799901,
(Marbaix, 2024a).

## 6 Authors contributions

P.M. proposed the concept, helped by previous discussions involving Z.Z. and A.M. P.M. and A.M designed the basis of the study. All contributed to identify missing information and links with relevant studies, and to improve the consistency and relevance of the manuscript. P.M. developed the database structure and collected most of its content, with contributions from
Z.Z. (risks related to bioenergy) and V.M. (risks in Europe). P.T led the developments related to temperature metrics (Sect. 2.2 and the related box). A.M led the development of Sect. 4.2 on further exploring the adaptation potential. V.M. contributed





to discussions and content on adaptation and on the challenges faced when assessing embers. P.M led the development of figures. P.T largely contributed to editing, with a critical eye on all aspects of the manuscript. All contributed to the introduction and to the concluding discussion on potential improvements and developments in future reports.

**7    Competing interests**

The contact author has declared that none of the authors has any competing interests.

**8    Acknowledgements**

Peter Thorne was supported by Co-Centre award number 22/CC/11103. The Co-Centre award is managed by Science Foundation Ireland (SFI), Northern Ireland's Department of Agriculture, Environment and Rural Affairs (DAERA) and UK
Research and Innovation (UKRI), and supported via UK's International Science Partnerships Fund (ISPF), and the Irish Government's Shared Island initiative.

We thank Professor Jean-Pascal van Ypersele (UCLouvain, Belgium) for supporting the concept when this research started.

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
