# Peer review of "Climate change risks illustrated by the IPCC "burning embers" Philippe Marbaix1, Alexandre K. Magnan2,3,4, Veruska Muccione5,6, Peter W. Thorne7, Zinta Zommers8"

_Earth System Science Data, 2024_

## Author Comment (AC2)

The development of a database on knowledge and assumptions underlying the IPCC burning embers is much needed and is likely to prove useful in understanding the evolution of risk judgments over time and in providing a critical foundation for future judgments and their comparability with previous embers. The illustrative applications of the associated climate risks ember explorer are effective. They show how analysis can use the by-now large number of embers that have been produced to draw broader conclusions about the balance of risks at different warming levels, the risk reduction possible from adaptation, and the types of risks that are relatively more or less serious.

I have no major reservations or suggestions for major revisions for this manuscript. I also find it clear and well organized.

> *We thank you for your positive evaluation of this work and your support for its future use.*

I have two broader suggestions, and then a number of more minor comments that I have listed below, in the order they appear.

The first broader comment is that the aggregation of risks across embers is useful but of course is subject to the distribution of the types of risks considered in the embers (some types of risks may be over- or under-represented). It would be useful to show early in the paper the distribution of risks considered. For example they could each be assigned to one of the Representative Key Risk (RKR) categories defined in the AR6 WG2 Ch 19, and the number of risks by category displayed in a figure. A similar figure could be made for risks by world region. Some of this categorization occurs in figures 7 and 8, but for a different purpose and it comes late in the paper.

> *Thank you for the suggestion. We agree that referring to the RKR categories adds relevant information from AR6. As a result, we attributed a "main category" to all embers except for the Reasons for Concern (RFCs). This can be viewed at https://climrisk.org/cree/list?keywords. To account for the fact that the scope of some of the embers is not entirely covered by a single category, we added an additional category when needed (and the abbreviation "RKR+" when it was not clear that the selected categories entirely cover the scope of the ember).*

*The description of the categories, based on AR6 WGII Table 16.6 supplemented with specific information as needed to assign a category to each ember, is as follows:*

| Code | Category | Description |
|---|---|---|
| RKR-A (coastal) | Low-lying coastal socio-ecological systems | Risks to ecosystem services, people, livelihoods and key infrastructure in low-lying coastal areas, associated with a wide range of hazards, including sea level changes, ocean warming and acidification, weather extremes (storms, cyclones), sea ice loss, etc. (16.5.2.3.1) *Note: We included risks to ecosystems themselves (even when the ember did not explicitly refer to ecosystem services) because 16.5.2.3.1 refers to « ecological and human components »; however human aspects tend to dominate the description of this category.* |
| RKR-B (ecosystems) | Terrestrial and ocean ecosystems | Transformation of terrestrial and ocean/coastal ecosystems, including change in structure and/or functioning, and/or loss of biodiversity (16.5.2.3.2) *Notes: - We included ecosystem services, based on AR6 WGII section 16.5.2.3.2. Carbon loss is also explicitly included. However, for food production-related embers, we used RKR-F as the main category, adding RKR-B as the second category. - Embers in "RKR-B" are only those which are not included in the two sub-categories below (this means that the ember does not specifically relate to land or to ocean systems)* |
| RKR-B.T | Terrestrial ecosystems | |
| RKR-B.O | Ocean ecosystems | |
| RKR-C (infrastructure) | Critical physical infrastructure, networks and services | Systemic risks due to extreme events leading to the breakdown of physical infrastructure and networks providing critical goods and services. |
| RKR-D (living standards) | Living standards | Economic impacts across scales, including impacts on gross domestic product (GDP), poverty and livelihoods, as well as the exacerbating effects of impacts on socioeconomic inequality between and within countries (16.5.2.3.4). *Note: This category was attributed to risks for tourism, considering that tourism is part of economies. However the focus of this category is on poverty and livelihoods, with no explicit reference to tourism in the description provided in AR6; we added "RKR+" (see below) to indicate that inclusion of tourism-related embers in RKR-D might be reconsidered in the future.* |
| RKR-E (health) | Human health | Human mortality and morbidity, including heat-related impacts and vector-borne and waterborne diseases (16.5.2.3.5) |
| RKR-F (food) | Food security | Food insecurity and the breakdown of food systems due to climate change effects on land or ocean resources (16.5.2.3.6) |
| RKR-G (water) | Water security | Risk from water-related hazards (floods and droughts and related disasters) and water quality deterioration, including water scarcity and risk to indigenous and traditional cultures and ways of life (16.5.2.3.7) |
| RKR-H (peace) | Peace and human mobility | Risks to peace within and among societies from armed conflict as well as risks to low-agency human mobility within and across state borders, including the potential for involuntarily immobile populations. (16.5.2.3.8) |
| RKR+ | More than 2 RKR categories or not clearly within the defined categories | Risks which span more than 2 AR6 RKR categories or that do not clearly fall within the defined categories: this would benefit from specific attention when further defining categories |

*We used this categorisation to provide a new overview of the risk categories covered in each chapter, taking all embers into account except for the RFCs:*

| RKR category | AR5-SYR | SR1.5-Chapter 3 | SROCC-Chapter 5 | SRCCL-Chapter 7 | AR6-WGII-CCP4 (Mediterranean) | AR6-WGII-CCP6 (Polar) | AR6-WGII-Chapter 2 (Terrestrial ecosystems) | AR6-WGII-Chapter 7 (Health) | AR6-WGII-Chapter 9 (Africa) | AR6-WGII-Chapter 11 (Australasia) | AR6-WGII-Chapter 13 (Europe) | AR6-WGII-Chapter (Water scarcity) | All chapters |
|---|---|---|---|---|---|---|---|---|---|---|---|---|---|
| RKR-A | 1 | 3 | | | 2 | 1 | | | | 2 | 3 | | 12 |
| RKR-B | | 1 | 5 | | | 4 | | | 1 | | | | 11 |
| RKR-B.O | 2 | 4 | 9 | | 1 | 2 | | | | 4 | 1 | | 23 |
| RKR-B.T | 2 | 1 | | 4 | 2 | | 5 | | | 4 | 1 | 1 | 20 |
| RKR-C | | | | 1 | 1 | | | | | 2 | | 2 | 6 |
| RKR-D | | 1 | | 1 | | | | | | | | 10 | 12 |
| RKR-E | | 1 | | | | 1 | | 18 | 1 | 2 | 2 | | 25 |
| RKR-F | | 4 | | 6 | 1 | 2 | | | 1 | 2 | 2 | 2 | 20 |
| RKR-G | | 1 | | 3 | 1 | | | | | | 6 | 4 | 15 |
| RKR+ | | | | | | 1 | | | | 2 | | | 3 |
| All RKRs | 5 | 16 | 14 | 15 | 8 | 11 | 5 | 18 | 3 | 18 | 15 | 19 | 147 |

Green : chapters focusing on regions

*This categorisation will form the basis for a revised version of table 3, providing a better overview of the distribution of risks early in the paper. This information will also be included in a revised version of figure 7.*

*We believe that this categorisation helps in illustrating which risks were assessed in which context (in particular, for which regional chapter in AR6). However, we would like to caution against (over-)interpreting the number of embers in a category as suggesting that a given 'type of risk' is better represented than others: it gives an indication, but a high number of embers may correspond to a highly disaggregated presentation of risks in a situation where the assessment is not deeper or broader than for other risks which were presented with fewer embers.*

*This categorisation also provides some information on the coverage of regions (highlighted in green in the above table). Our impression is that it would be hard, if not impossible, to provide more categorisation of embers within regions (except for two embers about the Arctic provided in the special reports, as indicated in Figure 8). Beyond that, one would need to find out which regions are considered in (or relevant to) the assessment for coral reefs, lyme disease, etc. Such a categorisation might be considered in future reports: authors may select relevant regions when assessing risks. If done in a systematic way, this could add relevant geographical information.*

The second comment is that I find the discussion section to be somewhat long and delving a bit further than necessary into topics that are related but not central to this paper. That includes some of the discussion of adaptation framing, limits to adaptation, and the final section on the future of burning embers. All of these sections are relevant to some extent, however, so there are arguments to keep them. The authors might consider ways to make them somewhat more concise.

*We have actively considered shortening some aspects in the revised manuscript. However, as you note, there is nothing irrelevant. Explaining the limitations that we learned and the questions that we faced while analysing the existing embers is important for future developments. Aspects which appear to be "detail" may not be the same in different contexts or for different experts. Also taking into account the comments of the other Referee, we will try to balance length and detail.*

*Minor comments:*

line 15: "due to the colours used" -> "with risk judgments reflected by the colours used"

*The sentence was revised.*

line 18-19: it should be specified what time period the database covers, that is, all embers created since the TAR

*This is now clarified.*

line 47: "temperature" -> "global average temperature"

*Done, thank you.*

line 49-55: It is probably worth mentioning here that the burning embers diagram did not appear in the IPCC Fourth Assessment Report, although the Reasons for Concern were re-assessed in the text. Instead, the BE diagram appeared in a paper (Smith et al.) that appeared in PNAS after the report was published. This is mentioned later in the paper, but it seems also appropriate here.

*This was missing and has been added, thank you for noticing.*

line 51: "with four discrete risk levels" -> "with four (rather than three) discrete risk levels"

*This was clarified.*

line 61-62: The shift from "key vulnerabilities" in AR4 to "key risks" in AR5 was not only a matter of a change in terminology. AR4 was somewhat murky in the distinction between vulnerability and risk, sometimes explicitly acknowledging their differences while at other times seeming to substitute one for the other. AR5 clarified the conceptual framework for risk, focusing the RFC assessment appropriately on risk as the ultimate outcome of interest. I suggest changing the wording from "which were later referred to as "key risks"" to "; AR5 refocused this approach on 'key risks'".

*We agree that it is not solely a change in wording and adapted the text accordingly. It is also important to provide the information that the concept of "criteria for key risks" was pioneered in AR4 (especially WGII section 19.2). Then the Special Report on Managing the risks of extreme events (...) changed and clarified the terminology. As a result, AR5 starts from key vulnerability and builds key risks, based on the concept that vulnerability is a component of risk. AR6 further focused on key risks.*

line 85: "United Framework" -> "United Nations Framework"

*Done.*

line 88: by saying this paper "proposes to structure a database" it is unclear whether such a database has been created yet; this should be stated more definitively.

*Done.*

line 98: the wording "quantitative estimates of how risk increases with the level of climate change" can be interpreted as estimates from a variety of studies of impacts relevant to a particular ember, which underlie the expert judgments made (eg, various estimates of the damages from flooding that might underlie the RFC related to extreme events). Since this is not what you intend, I suggest changing to "quantitative estimates of the global average temperature at which the risk for a given ember changes from one level to another".

*Done (we agree that your wording is more precise).*

line 172-175: This is a useful finding: "While getting the numerical data to reproduce the embers has become easier in the recent IPCC reports, it remains difficult to get a synthetic description of the risks illustrated in each ember and an explanation for each risk transition. This information is rarely associated with the quantitative data and was not always collected in a systematic way."

*We agree, guidelines and processes to help in further improving this in the next reports would be helpful.*

line 256: Section 2.3 on Adaptation levels and scenarios: This section describes well the difficulties of conflating adaptation assumptions with SSP scenarios (which do not include

adaptation by definition), which the database appears to do by lumping them into the same field of information. However the section does not present a solution to the challenge described. I don't think there is a good reason to record adaptation levels and scenario in the same field; it excludes for example the possibility of having an SSP3 with high adaptation vs an SSP3 with low adaptation. A better solution here is needed; I suggest the scenario and adaptation assumptions should be separated.

> *We believe that the current solution fills the needs of embers produced so far. An entry in the "scenario" table (figure 2) does not necessarily refer to an SSP or adaptation level; "scenario" can be understood as a context for the ember. Technically, the scenario table could contain an SSP3 with low adaptation and an SSP3 with high adaptation. However, to date, when two or more levels of adaptation were considered for risks represented by an ember, each was linked to a single SSP (as discussed in the manuscript). If different adaptation levels are considered for a given SSPs in future studies or reports, the existing framework can accommodate for this, as a first step. If an additional field is attached to each ember (e.g. to separately assign an adaptation level), we would suggest taking care that the new field is as independent as possible from the others, while each of its values accurately characterises an element of the context shared by several embers. This would maximise the usefulness of the new field in the structure of the database. It might be that this field would relate to vulnerability and/or exposure instead of adaptation, focusing on the risk factor instead of the magnitude of adaptation effort (similarly to SSPx-y, with 'y' a radiative forcing level: the 'y' relates to the magnitude of the effect on climate, not the amount of mitigation efforts, which is a consequence of the combination of x and y; here vulnerability and exposure might be 'y' and the adaptation would be what is needed to get there given the context set by 'x'). As such situations with multiple adaptation levels for the same SSP were not considered within AR6 embers, we prefer delaying further discussion to if and when the situation is considered in a future IPCC assessment product and only then supplement the database accordingly (the database is managed in a standard way, which makes such adaptations easy).*

p. 15: Table 3: the caption should include a clear and comprehensive description of the color scheme used for shading rows.

> *Well noted. This table will be entirely revised (see response on RKR-categories), also taking into account that colour shadings are not allowed in tables.*

line 353: Section 2.5.3: an excellent plan for collaboration on completing the information in the database.

> *Thank you*

line 471: in this figure caption, I found the explanation of panel (c) hard to understand. I suggest changing "indicates the fraction of assessed embers, at each GMT, for which the risk is above the midpoint within each transition" to: "indicates the fraction of assessed embers for which a given GMT exceeds the midpoint of each of three risk transitions"

*We agree that this wording is clearer, thank you.*

p. 26: Table 4 is quite useful in illustrating how the assessment of the database of embers can yield useful information about risks at the low or high end of the distribution.

*Thank you*

p. 33-34: I am not sure figures 7 and 8 work very well. They are very hard to understand, particularly the lines that connect different results across adaptation levels. I believe that these are supposed to be interpreted that each line represents a separate result: what is the change in risk at the same warming level but with different adaptation assumptions. However since many of the risk judgments overlap, there are multiple lines that all look like they are connected to each other and one doesn't know what to do with that. Maybe each line could be made into an arrow, so that they appeared more separate than connected?

*Thanks for this remark. We agree that the lines appear more connected than they actually are, because lines connecting different GMT levels have been drawn in the same way. We are considering improvements to this design, including by changing the thickness of the lines in connection with the GMT levels, similarly to how the size of circles correspond to GMT levels. Beyond that, those figures are the result of multiple attempts to illustrate the set of embers. It is a hard task, but we think that the figures are innovative and hope that they can provide a basis for thinking about even better synthesis figures in the future.*

---

## Author Response (AR1)

*Climate change risks illustrated by the IPCC "burning embers" (essd-2024-312):*

**Author's response to comments**

Two referees made comments on the manuscript. This file contains the responses, comment by comment, with the following structure:

- The comment is in black,
- The response is in blue indented text,
- Further information about changes to the manuscript is in green indented text

The responses are identical to those given in the framework of the online discussion, with minor restructuring of the text to highlight the changes in the manuscript.

**Referee 1**

[General comments]

The paper overviews the history of "burning embers" comprehensively in the Introduction chapter, explains the structure of database for archiving knowledge on climate risks and communicating them with the burning-embers format in Chapter 2, exemplifies analyses in Chapter 3, and finally discuss possible contributions of the database to future risk assessments and update of the burning-embers.

I highly evaluate this paper with the following reasons in summary and hope that it is published as a reviewed article on ESSD to be read by a wide range of readers.

Timely article for the initial period of the IPCC-AR7 cycle: Just at the initial period of the IPCC-AR7 cycle, this article will be beneficial both for researchers contributing to the assessment report as a lead author of WG2 and researchers who are willing to conduct research to be assessed properly in the report. Traceability and objectivity of the burning embers assessment have been strengthened gradually for the previous 20 years. This paper will significantly contribute to the further improvement of the RFCs and burning embers approach both from theoretical and practical aspects. Things discussed in Section 4 are describing current research gaps concisely and will send useful signals to impact projection researchers who are willing to contribute to the IPCC-AR7's risk assessment. Researchers may also use this paper for explaining the potential value of their new research proposal to funders in the coming years.

> *We thank you for this positive evaluation of our manuscript.*

Potential flexibility of the proposed database structure: We are not sure how long the proposed database continues to work effectively. Key aspects or uncertainties of risk analyses may radically change in future and database for storing analyses outputs will need to be flexibly revised or extended to be continuously functional. The authors of the paper seem conscious about it and they are not selling the current design of the database as the ultimate and perfect one. I suppose the attitude will allow effective extension and improvement of the database structure in future.

*This is our wish and intention. As a rule, having data well-structured while avoiding the introduction of more details and/or structural elements than needed can be expected to facilitate future changes, which will need to be discussed with researchers assessing or synthesising impacts.*

Well balanced technical documentation: This paper not only explains the technical detail of the database structure but also exemplifies how the database can be really used for storing and communicating climate risk assessments outputs in Chapter 3, that would help readers contribute to the community effort for fulfilling risk analyses.

[Specific technical suggestions]

Table 5 (P28): From the viewpoint of decimal position, "2" in some cells should be written as "2.0".

> *Thanks, this is corrected.*

4.2.1 (P39): There is no 4.2.2 to be put in parallel here. Considering the logical flow and structure of the story, it may not be needed to be separately put as 4.2.1 but connected to the previous paragraphs (as a part of 4.2).

> *We agree that there was a problem, thank you. Our perception is that it is useful to have two subtitles in section 4.2 to clarify the structure.*
>
> *Changes to the manuscript:*
>
> *We have added a new subsection numbered 4.2.1, which includes the (existing) content on adaptation potential, and the existing subtitle becomes 4.2.2, with a focus on the limits of adaptation.*

**Referee 2**

The development of a database on knowledge and assumptions underlying the IPCC burning embers is much needed and is likely to prove useful in understanding the evolution of risk judgments over time and in providing a critical foundation for future judgments and their comparability with previous embers. The illustrative applications of the associated climate risks ember explorer are effective. They show how analysis can use the by-now large number of embers that have been produced to draw broader conclusions about the balance of risks at different warming levels, the risk reduction possible from adaptation, and the types of risks that are relatively more or less serious.

I have no major reservations or suggestions for major revisions for this manuscript. I also find it clear and well organized.

> *We thank you for your positive evaluation of this work and your support for its future use.*

I have two broader suggestions, and then a number of more minor comments that I have listed below, in the order they appear.

The first broader comment is that the aggregation of risks across embers is useful but of course is subject to the distribution of the types of risks considered in the embers (some types of risks may be over- or under-represented). It would be useful to show early in the paper the distribution of risks considered. For example they could each be assigned to one of the Representative Key Risk (RKR) categories defined in the AR6 WG2 Ch 19, and the number of risks by category displayed in a figure. A similar figure could be made for risks by world region. Some of this categorization occurs in figures 7 and 8, but for a different purpose and it comes late in the paper.

*Thank you for the suggestion. We agree that referring to the RKR categories adds relevant information from AR6. As a result, we attributed a "main RKR category" to all embers except for the Reasons for Concern (RFCs). General information on the RKR is provided in the revised table 3 of the manuscript, and the details can be viewed at https://climrisk.org/cree/list?keywords. To account for the fact that the scope of some of the embers is not entirely covered by a single category, we added an additional category when needed. In addition, when the selected categories do not clearly cover the full scope of the ember, we added the specific abbreviation "RKR-X" (the abbreviation was 'RKR+' in our online response).*

*We believe that this categorisation helps in illustrating which risks were assessed in which context (in particular, for which regional chapter in AR6). However, we would like to caution against (over-)interpreting the number of embers in a category as suggesting that a given 'type of risk' is better represented than others: it gives an indication, but a high number of embers may correspond to a highly disaggregated presentation of risks in a situation where the assessment is not deeper or broader than for other risks which were presented with fewer embers.*

*The categorisation in the revised table 4 provides some information on the coverage of regions. Our impression is that it would be hard, if not impossible, to provide more categorisation of embers within regions (except for two embers about the Arctic provided in the special reports, as indicated in Figure 8). Beyond that, one would need to find out which regions are considered in (or relevant to) the assessment for coral reefs, lyme disease, etc. Such a categorisation might be considered in future reports: authors may select relevant regions when assessing risks. If done in a systematic way, this could add relevant geographical information.*

*Changes to the manuscript:*

*The RKR categorisation forms the basis for the revised version of table 3, providing a better overview of the distribution of risks early in the paper. Table 3 includes a description of the RKRs, based on AR6 WGII Table 16.6 supplemented with specific information (indicated in italics in the table) as needed to assign a category to each ember. This categorisation is used in table 4 to provide an overview of embers within each chapter and RKR cluster. The text of section 2.4 is adapted to reflect these changes and the above explanation.*

*Figure 7 is changed in a corresponding way, on the basis of the identified RKRs.*

The second comment is that I find the discussion section to be somewhat long and delving a bit further than necessary into topics that are related but not central to this paper. That includes

some of the discussion of adaptation framing, limits to adaptation, and the final section on the future of burning embers. All of these sections are relevant to some extent, however, so there are arguments to keep them. The authors might consider ways to make them somewhat more concise.

> *We have actively considered shortening some aspects in the revised manuscript. However, as you note, there is nothing irrelevant. Explaining the limitations that we learned and the questions that we faced while analysing the existing embers is important for future developments. Aspects which appear to be "detail" may not be the same in different contexts or for different experts. Also taking into account the comments of the other Referee, we tried to balance length and detail.*

> *Changes to the manuscript: a few sentences were removed or shortened, but without a net length reduction because other clarifications were needed.*

*Minor comments:*

line 15: "due to the colours used" -> "with risk judgments reflected by the colours used"

> *The sentence was revised.*

line 18-19: it should be specified what time period the database covers, that is, all embers created since the TAR

> *This is now clarified.*

line 47: "temperature" -> "global average temperature"

> *Done, thank you.*

line 49-55: It is probably worth mentioning here that the burning embers diagram did not appear in the IPCC Fourth Assessment Report, although the Reasons for Concern were re-assessed in the text. Instead, the BE diagram appeared in a paper (Smith et al.) that appeared in PNAS after the report was published. This is mentioned later in the paper, but it seems also appropriate here.

> *This was missing and has been added, thank you for noticing.*

line 51: "with four discrete risk levels" -> "with four (rather than three) discrete risk levels"

> *This was clarified.*

line 61-62: The shift from "key vulnerabilities" in AR4 to "key risks" in AR5 was not only a matter of a change in terminology. AR4 was somewhat murky in the distinction between vulnerability and risk, sometimes explicitly acknowledging their differences while at other times seeming to substitute one for the other. AR5 clarified the conceptual framework for risk, focusing the RFC assessment appropriately on risk as the ultimate outcome of interest. I suggest changing the wording from "which were later referred to as "key risks"" to "; AR5 refocused this approach on 'key risks'".

*We agree that it is not solely a change in wording.  It is also important to provide the information that the concept of "criteria for key risks" was pioneered in AR4 (especially WGII section 19.2). Then the Special Report on Managing the risks of extreme events (...) changed and clarified the terminology. As a result, AR5 starts from key vulnerability and builds key risks, based on the concept that vulnerability is a component of risk. AR6 further focused on key risks.*

*Changes to the manuscript:*

*The related sentence, on page 3, lines 76-77, was revised to clarify that changes between AR4 and AR5 include "changes and clarifications in the conceptualization of vulnerability and risk".*

Changes to the manuscript :

line 85: "United Framework" -> "United Nations Framework"

*Done.*

line 88: by saying this paper "proposes to structure a database" it is unclear whether such a database has been created yet; this should be stated more definitively.

*Done.*

line 98: the wording "quantitative estimates of how risk increases with the level of climate change" can be interpreted as estimates from a variety of studies of impacts relevant to a particular ember, which underlie the expert judgments made (eg, various estimates of the damages from flooding that might underlie the RFC related to extreme events). Since this is not what you intend, I suggest changing to "quantitative estimates of the global average temperature at which the risk for a given ember changes from one level to another".

*Done (we agree that your wording is more precise).*

line 172-175: This is a useful finding: "While getting the numerical data to reproduce the embers has become easier in the recent IPCC reports, it remains difficult to get a synthetic description of the risks illustrated in each ember and an explanation for each risk transition. This information is rarely associated with the quantitative data and was not always collected in a systematic way."

*We agree, guidelines and processes to help in further improving this in the next reports would be helpful.*

line 256: Section 2.3 on Adaptation levels and scenarios: This section describes well the difficulties of conflating adaptation assumptions with SSP scenarios (which do not include adaptation by definition), which the database appears to do by lumping them into the same field of information. However the section does not present a solution to the challenge described. I don't think there is a good reason to record adaptation levels and scenario in the same field; it excludes for example the possibility of having an SSP3 with high adaptation vs an SSP3 with low adaptation. A better solution here is needed; I suggest the scenario and adaptation assumptions should be separated.

*We believe that the current solution fills the needs of embers produced so far. An entry in the "scenario" table (figure 2) does not necessarily refer to an SSP or adaptation level; "scenario" can be understood as a context for the ember. Technically, the scenario table could contain an SSP3 with low adaptation and an SSP3 with high adaptation. However, to date, when two or more levels of adaptation were considered for risks represented by an ember, each was linked to a single SSP (as discussed in the manuscript). If different adaptation levels are considered for a given SSPs in future studies or reports, the existing framework can accommodate for this, as a first step. If an additional field is attached to each ember (e.g. to separately assign an adaptation level), we would suggest taking care that the new field is as independent as possible from the others, while each of its values accurately characterises an element of the context shared by several embers. This would maximise the usefulness of the new field in the structure of the database. It might be that this field would relate to vulnerability and/or exposure instead of adaptation, focusing on the risk factor instead of the magnitude of adaptation effort (similarly to SSPx-y, with 'y' a radiative forcing level: the 'y' relates to the magnitude of the effect on climate, not the amount of mitigation efforts, which is a consequence of the combination of x and y; here vulnerability and exposure might be 'y' and the adaptation would be what is needed to get there given the context set by 'x').*

*Changes to the manuscript:*

*There are no changes to the manuscript specifically addressing this comment. We hope that the above clarification will be accepted by the Referees. The key point is that we do not reject the possibility of cases such as an "SSP3 with high adaptation vs an SSP3 with low adaptation". However, such situations with multiple adaptation levels for the same SSP were not considered within AR6 embers. Therefore, we prefer delaying further discussion to if and when the situation is considered in a future IPCC assessment product and only then supplement the database accordingly (the database is managed in a standard way, which makes such adaptations easy).*

p. 15: Table 3: the caption should include a clear and comprehensive description of the color scheme used for shading rows.

*Well noted.*

*This table has been entirely revised (see response on RKR-categories).*

line 353: Section 2.5.3: an excellent plan for collaboration on completing the information in the database.

*Thank you*

line 471: in this figure caption, I found the explanation of panel (c) hard to understand. I suggest changing "indicates the fraction of assessed embers, at each GMT, for which the risk is above the midpoint within each transition" to: "indicates the fraction of assessed embers for which a given GMT exceeds the midpoint of each of three risk transitions"

*We agree that this wording is clearer, thank you.*

*The change was done as suggested by the Referee.*

p. 26: Table 4 is quite useful in illustrating how the assessment of the database of embers can yield useful information about risks at the low or high end of the distribution.

*Thank you*

p. 33-34: I am not sure figures 7 and 8 work very well. They are very hard to understand, particularly the lines that connect different results across adaptation levels. I believe that these are supposed to be interpreted that each line represents a separate result: what is the change in risk at the same warming level but with different adaptation assumptions. However since many of the risk judgments overlap, there are multiple lines that all look like they are connected to each other and one doesn't know what to do with that. Maybe each line could be made into an arrow, so that they appeared more separate than connected?

*Thanks for this remark. We agree that the lines appeared more connected than they actually are, because lines connecting different GMT levels have been drawn in the same way. Beyond that, those figures are the result of multiple attempts to illustrate the set of embers. It is a hard task, but we think that the figures are innovative and hope that they can provide a basis for thinking about even better synthesis figures in the future.*

*Changes to the manuscript:*

*Figure 7 and 8 have been revised to cluster the risks by RKR and indicate these categories. We changed the thickness of the connecting lines related to the GMT levels, similarly to how the size of circles corresponds to GMT levels.*